# UNPAIRED SINGLE-CELL DATASET ALIGNMENT WITH WAVELET OPTIMAL TRANSPORT

## ABSTRACT

Aligning single-cell samples across different datasets and modalities is an important task with the rise of high-throughput single-cell technologies. Currently, collecting multi-modality datasets with paired samples is difficult, expensive, and impossible in some cases, motivating methods to align unpaired samples from distinct uni-modality datasets. While dataset alignment problems have been addressed in various domains, single-cell data introduce additional complexity including high levels of noise, dropout, and non-isometry between data spaces. In response to these unique challenges, we propose *Wavelet Optimal Transport* (WOT), a multi-resolution optimal transport method that aligns samples by minimizing the *spectral graph wavelet* discrepancies across datasets. Filters are incorporated into the optimization process to eliminate non-essential scales and wavelets, enhancing the quality of correspondences. We demonstrate the capacity of WOT in highly noisy and non-isometric conditions, outperforming previous state-of-the-art methods by significant margins, especially on real single-cell datasets.

## 1 INTRODUCTION

Single-cell technologies have revolutionized biological research by offering a detailed understanding of individual cell behaviors within heterogeneous populations. However, most single-cell technology is only capable of capturing a single description of the cell state (Trapnell, 2015); experiments such as single-cell proteomics or Western blot can even be *destructive*, meaning they either alter or destroy the cell being analyzed, preventing further analysis in the same cell (Tang, 2022). Thus, an increasingly essential but difficult problem within this field is aligning the data produced by each of these technologies in which no paired samples are available.

The problem of data or manifold alignment is not unique to biology and has been extensively studied elsewhere. Works in natural language processing have aligned the data spaces of different languages (Alvarez-Melis & Jaakkola, 2018; Schuster et al., 2019; Vulić et al., 2019), and the field of computer vision has translated images between different domains (Zhu et al., 2017; Grover et al., 2020; Su et al., 2022). In addition to the strict absence of paired data, the alignment of single-cell data poses additional unique issues that are often not present in other fields. For single-cell data, identifying correspondences between datasets requires navigating the inherent modality-specific variability and noise they present. Although such variability can be attributed to biological variabilities like cell cycle stages, spatial heterogeneity, and cellular differentiation, it can also be caused by technical variabilities like dropout, batch effect, and library preparation (Arzalluz-Luque et al., 2017). Thus, proposed methods in this field must be able to identify correspondences based on the important biological variability while filtering out the unimportant, technical variability—all in a completely unpaired setting.

To address the challenges of unpaired single-cell alignment, we propose *Wavelet Optimal Transport* (WOT), a framework that finds a transport plan that agrees with multiple views of a dataset while filtering out uninformative or noisy components. Specifically, our framework considers the relationships of samples in a dataset as the coefficients of *spectral graph wavelets*, allowing us to decompose the dataset's signals into both scale and individual sample resolution. WOT aligns points between datasets such that it minimizes the discrepancy between the wavelets of each dataset across all views. We incorporate a *filter* in the optimization of the transport plan that removes uninfor-

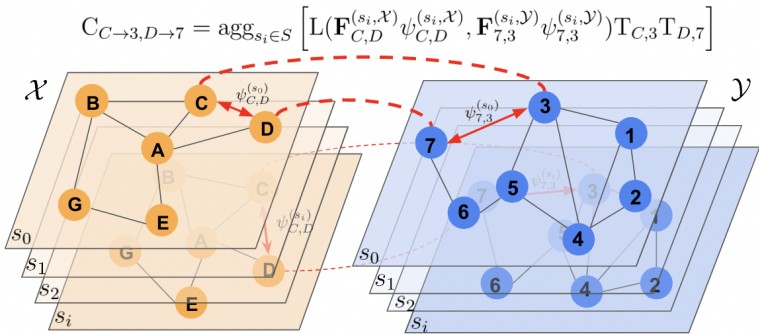

$$C_{C\to 3,D\to 7} = \text{agg}_{s_i \in S} \left[ L(\mathbf{F}_{C,D}^{(s_i,\mathcal{X})} \psi_{C,D}^{(s_i,\mathcal{X})}, \mathbf{F}_{7,3}^{(s_i,\mathcal{Y})} \psi_{7,3}^{(s_i,\mathcal{Y})}) T_{C,3} T_{D,7} \right]$$

Figure 1: **Overview of the Wavelet Optimal Transport (WOT) Framework**. Given two graphs $\mathcal{X}$ and $\mathcal{Y}$, the intra-graph relationships of nodes are defined by their *spectral graph wavelet coefficients* ($\psi$) such that we view the graph under multiple scales or frequencies $S$. WOT finds a transport plan T that minimizes the total cost C, which *aggregates* across all scales. One component of this cost is $C_{C\to 3,D\to 7}$, representing the aggregated cost when nodes C and D in $\mathcal{X}$ are mapped to the respective nodes 3 and 7 in $\mathcal{Y}$. The *filter* $\mathbf{F}$ removes uninformative scales and wavelets.

mative frequencies or regions of the data, providing more robust matches across datasets. Figure 1 summarizes each of these components in the WOT framework.

We provide two implementations of WOT, entropy-based WOT (E-WOT) and learned WOT (L-WOT), each with its own specification of the filter. E-WOT uses the entropy of each scale as the filter. L-WOT, on the other hand, jointly *learns* the filter with the transport plan in an alternating minimization-maximization strategy. Empirically, we demonstrate the effectiveness of E-WOT and L-WOT in aligning highly noisy datasets, aligning points between non-isometric shapes, and ultimately aligning samples in two real single-cell multi-omics datasets.

To summarize, our core contributions are:

1. We formulate the relationships of points in each dataset in terms of spectral graph wavelet coefficients, and develop a multi-scale optimal transport framework that finds a transport plan between points in each dataset with respect to these wavelets.
2. We provide two implementations of our framework, E-WOT and L-WOT, that allow us to filter out unimportant scales and wavelets, elucidating the important structures to match between datasets.
3. We demonstrate that WOT is more robust to noise, dropout, and non-isometry—advantages that enable us to consistently outperform previous state-of-the-art methods in aligning single-cell multi-omics datasets, scGEM and SNARE-seq, in an entirely unpaired setting.

## 2 BACKGROUND

### 2.1 SINGLE-CELL DATASET ALIGNMENT

Several methods have been proposed to align single-cell datasets. SCOT (Demetci et al., 2022b) and its updated version, SCOTv2 (Demetci et al., 2022a), use the Gromov-Wasserstein distance and its unbalanced formulations to align distributions across different domains. Another approach, Pamona (Cao et al., 2022), leverages a combination of manifold learning and partial Gromov-Wasserstein distance. UnionCom (Cao et al., 2020) integrates datasets by preserving both individual and shared structures through a shared low-dimensional embedding. MMD-MA (Liu et al., 2019) is based on maximum mean discrepancy, offering a non-parametric way to integrate single-cell data. Lastly, cross autoencoders (Yang et al., 2021) utilize autoencoders to learn common embeddings that can bridge the gap between different modalities. While these works have shown success in finding high-quality correspondences given prior knowledge of cell types or a subset of paired samples, they (1) still often underperform when aligning datasets in a completely *unpaired* setting and (2) do not

explicitly reduce dataset-intrinsic noise or signal (which is later shown in Section 4.3 to be important for accurate alignment across single-cell datasets).

## 2.2 Optimal Transport and Gromov-Wasserstein Distance

Optimal transport (OT) (Villani et al., 2009) is an appealing solution to the data alignment problem. The goal of OT is to find the best way to transform one distribution into another. However, when dealing with different spaces (e.g. ATAC-seq space vs nuclei imaging space), a direct comparison becomes challenging. Using the Gromov-Wasserstein (GW) distance (Mémoli, 2011) provides a framework that bypasses this issue and has become a popular choice for many data alignment tasks (Gong et al., 2022; Thual et al., 2022; Li et al., 2022). Instead of comparing points directly by their *positions* across datasets, OT based on the Gromov-Wasserstein distance compares how the *distance matrices* of points are mapped across different domains.

Formally, given two datasets $A = \{\mathbf{a}_i\}_{i=1}^n$ and $B = \{\mathbf{b}_i\}_{i=1}^m$, consider two discrete metric spaces $(A, d_A)$ and $(B, d_B)$ with probability measures $\mathbf{p}$ and $\mathbf{q}$, respectively. In this setting, the Gromov-Wasserstein distance identifies a transport plan (or *coupling*) $\mathrm{T}^*$ among the set of joint distributions between $A$ and $B$ with marginals $\mathbf{p}$ and $\mathbf{q}$. Namely, $\mathrm{T}^* \in \Pi(\mathbf{p}, \mathbf{q}) = \{\mathrm{T} \in \mathbb{R}_{\geq 0}^{n \times m} : \mathrm{T}\mathbb{1}_m = \mathbf{p}, \mathrm{T}^\top \mathbb{1}_n = \mathbf{q}\}$ minimizes a loss function $\mathrm{L} : \mathbb{R} \times \mathbb{R} \to \mathbb{R}$ measuring the discrepancy between pairs of points in each dataset (Alvarez-Melis & Jaakkola, 2018; Mémoli, 2011):

$$\mathrm{GW}(\mathbf{p}, \mathbf{q}) = \inf_{\mathrm{T} \in \Pi(\mathbf{p}, \mathbf{q})} \sum_{i,k=1}^n \sum_{j,l=1}^m \mathrm{L}(d_A(\mathbf{a}_i, \mathbf{a}_k), d_B(\mathbf{b}_j, \mathbf{b}_l))\mathrm{T}_{ij}\mathrm{T}_{kl} \tag{1}$$

This formulation allows for a comparison of the structural (metric) differences between the two spaces without explicitly comparing individual points.

As the field has expanded, variants of Gromov-Wasserstein OT have emerged. The entropy-regularized version, for example, includes an entropy term for smoother, more computationally feasible solutions (Peyré et al., 2016). Unbalanced (Séjourné et al., 2021) and partial formulations (Chapel et al., 2020) have also been introduced. Many new works have leveraged Gromov-Wasserstein OT in applications including domain adaptation (Yan et al., 2018), generative modeling (Bunne et al., 2019), and shape matching (Mémoli, 2011). However, we later show that Gromov-Wasserstein OT often fails in high noise regimes or when the two spaces are significantly different in structure.

## 2.3 Spectral Graph Wavelets

Wavelets can be viewed as augmentations of the Fourier bases, providing resolution in both time (or space) and frequency. Hammond et al. (2011) extended wavelets to the domain of graphs with Spectral Graph Wavelets (SGWs), allowing localized signal representation on both the vertex and frequency domain. Utilizing the spectral characteristics of the graph Laplacian to analyze and process signals on graphs, SGWs have been applied widely in machine learning. In the context of networks, Donnat et al. (2018) demonstrated the application of SGWs for node embeddings, leveraging their ability to encapsulate structural network information. Another work (Mémoli, 2009) proposed a heat kernel-based Gromov-Wasserstein distance, drawing parallels to geodesic and diffusion distances. We build on SGWs to develop a flexible optimal transport framework that not only generalizes to other wavelets (outside of the heat kernel) but also provides systematic approaches to filtering out uninformative wavelets and frequency bands.

## 3 Wavelet Optimal Transport

### 3.1 Preliminaries

Given datasets $A = \{\mathbf{a}_i\}_{i=1}^n$ and $B = \{\mathbf{b}_i\}_{i=1}^m$, we frame *dataset alignment* as an optimal transport task that aims to find the coupling $\mathrm{T} \in \Pi(\mathbf{p}, \mathbf{q})$ between samples in each dataset. $\Pi(\mathbf{p}, \mathbf{q})$ denotes the set of all joint distributions (transport plans) with empirical marginals $\mathbf{p} \in [0, 1]^n$ and $\mathbf{q} \in [0, 1]^m$ defined over samples $\{\mathbf{a}_i\}_{i=1}^n$ and $\{\mathbf{b}_i\}_{i=1}^m$ such that $\sum_i^n \mathbf{p}_i = 1$ and $\sum_i^m \mathbf{q}_i = 1$.

To obtain the spectral graph wavelets for dataset $X \in \{A, B\}$, assume $X$ has a fully connected weighted graph with weighted adjacency matrix $W \in \mathbb{R}_+^{|X| \times |X|}$. Different choices of metrics can be used to compute the affinity $W_{ij}$ between nodes $i$ and $j$ (particular implementations based on the RBF-kernel and geodesic distance are detailed in Appendix A.1). Second, the normalized[1] graph Laplacian of $X$ is computed as $\mathcal{L} = I_{|X|} - D^{-\frac{1}{2}} W D^{-\frac{1}{2}}$ (with $D$ as the diagonal degree matrix and $I_{|X|}$ as the identity matrix) and eigendecomposed as $\mathcal{L} = \mathbf{U} \Lambda \mathbf{U}^{-1}$. Finally, the eigenvectors $\mathbf{U}_i$ and the eigenvalues $\lambda_i = \Lambda_{ii}$ are combined with a wavelet generating function $g : \mathbb{R}^+ \to \mathbb{R}^+$, satisfying $g(0) = 0$ and $\lim_{x \to \infty} g(x) = 0$, to obtain the wavelet coefficients at scale $s > 0$:

$$\psi_{ij}^{(s)} = \sum_k g(s\lambda_k) \mathbf{U}_{ik}^\top \mathbf{U}_{jk} \tag{2}$$

In practice, to avoid the cost of diagonalizing the graph Laplacian to compute Equation (2), we leverage the Chebyshev polynomial approximation proposed in Hammond et al. (2011).

Intuitively, the $j$-th column of the matrix $\psi^{(s,X)} \in \mathbb{R}^{|X| \times |X|}$ corresponds to the wavelet of node $j$ whereas the coefficient $\psi_{ij}^{(s)} \in \mathbb{R}$ can be interpreted as the impact that the signal on node $i$ has on node $j$. The kernel $g$ modulates the spectral bands of the signals: depending on $s$, $g$ may emphasize the eigenvectors corresponding to larger eigenvalues (i.e. ones that carry high-frequency signals) versus the eigenvectors corresponding to smaller eigenvalues (i.e. ones that carry low-frequency signals). Wavelets at higher scales capture more local structures of the graph, while wavelets at lower scales capture higher-level patterns of the graph.

## 3.2 THE FRAMEWORK

Gromov-Wasserstein OT incorporates the underlying geometry of each space as induced by their metric. Here, we interpret the *spectral graph wavelet coefficients*, $\psi^{(s,A)}$ and $\psi^{(s,B)}$, as our intra-space similarity metric, allowing us to view the relationship between samples on multiple scales. Our framework leverages this multi-scale approach to find matches between $A$ and $B$ that are more robust to noise and non-isometry, notably by either highlighting or suppressing wavelet coefficients at specific scales and by aggregating the loss function over various scales.

Concretely, consider a discrete set of chosen scales $S = \{s_i \in \mathbb{R}_+\}_{i=1}^{|S|}$ and the associated sets of spectral graph wavelet coefficients $\psi^A = \{\psi^{(s,A)}\}_{s \in S}$ and $\psi^B = \{\psi^{(s,B)}\}_{s \in S}$ for datasets $A$ and $B$, respectively. We define the *Wavelet Optimal Transport* distance as

$$\text{WOT}(\psi^A, \psi^B, \mathbf{p}, \mathbf{q}, \mathbf{F}^A, \mathbf{F}^B, S) = \inf_{\mathbf{T} \in \prod(\mathbf{p},\mathbf{q})} \mathbf{C}(\psi^A, \psi^B, \mathbf{F}^A, \mathbf{F}^B, S, \mathbf{T}) \tag{3}$$

$$\text{where} \quad \mathbf{C} = \sum_{i,k=1}^n \sum_{j,l=1}^m \text{agg}_{s \in S} \left[ \mathbf{L}\left( \mathbf{F}_{ik}^{(s,A)} \psi_{ik}^{(s,A)}, \mathbf{F}_{jl}^{(s,B)} \psi_{jl}^{(s,B)} \right) \mathbf{T}_{ij} \mathbf{T}_{kl} \right] \tag{4}$$

with agg $: \mathbb{R}^{|S|} \to \mathbb{R}$ as an aggregation operation over scales, $\mathbf{L} : \mathbb{R} \times \mathbb{R} \to \mathbb{R}$ as the discrepancy measure between pairs of points, and $\mathbf{F}^A \in \mathbb{R}_+^{|S| \times n \times n}$ and $\mathbf{F}^B \in \mathbb{R}_+^{|S| \times m \times m}$ as scale-specific filters that highlights or suppresses wavelet coefficients.

We provide details on optimizing this objective in Sections 3.2 and 3.3. Now, we discuss the three primary components of our framework: (1) the *filter*, (2) the *wavelet* coefficients, and (3) the *aggregation* scheme.

**Filter.** The goal of the filters, $\mathbf{F}^A$ and $\mathbf{F}^B$, is to emphasize salient scales and coefficients while discounting noisy scales and coefficients between the two graphs. While the filters can be continuous (i.e. $\mathbf{F}^X \in \mathbb{R}_{\geq 0}^{|S| \times |X| \times |X|}$), they can also be a binary mask (i.e. $\mathbf{F}^X \in \{0, 1\}^{|S| \times |X| \times |X|}$) to sparsity the set of coefficients—our proceeding implementations in Section 3.3 and Section 3.4 only refer to the continuous version.

In its most basic formulation, all scales and wavelets can be weighted equivalently by setting $\mathbf{F}^A = \mathbf{J}_n = \mathbb{1}_n \otimes \mathbb{1}_n^\top$ and $\mathbf{F}^B = \mathbf{J}_m = \mathbb{1}_m \otimes \mathbb{1}_m^\top$; we refer to this formulation as vanilla-WOT. Depending

---

[1]Here, the normalized graph Laplacian is preferred since we construct a fully connected weighted graph, but the unnormalized graph Laplacian ($\mathcal{L} = D - W$) can also be used given a different graph construction from the dataset.

on the application, if there exists prior knowledge about (i) specific *scales* or (ii) specific *coefficients* that are known to be important or noisy, this prior can be easily integrated into the WOT framework using $\mathbf{F}^A$ and $\mathbf{F}^B$. For example, if there is a dataset with similar assumptions to Deutsch et al. (2016) where high-frequency scale $s$ is considered noise, we can set $\mathbf{F}^{(s,X)} = \mathbf{0}$. However, if there is no prior knowledge, we resort to heuristic filters (Section 3.3) and learned filters (Section 3.4).

**Spectral Graph Wavelet Kernels.** Different choices of $g$ allow for different scaling behaviors. Low-pass kernels allow frequencies below a certain cutoff threshold, effectively smoothing out the high-frequency components. This particularly preserves the gross features of a graph signal. While low-pass kernels do not satisfy the original properties of wavelet generating functions (since $g(0) \neq 0$, violating the conditions specified in Section 3.1), they are still included in our analysis because of their effectiveness. Band-pass kernels are designed to allow a specific band or range of frequencies to pass through, thus providing a lens to discern localized features in the graph signal. Tight frame kernels (Chan et al., 2004) are a subset of band-pass kernels that conserve the energy of the signal during the wavelet transformation (and its inverse), allowing for more accurate signal representation.

In this work, our evaluation is limited to the following set of wavelet kernels: the low-pass heat kernel (Davies, 1989), a tight frame Meyer kernel (Leonardi & Van De Ville, 2011), and a simple tight frame kernel provided by Defferrard et al.. We provide a performance analysis of different kernels in Section 4 and the details of constructing spectral graph wavelets in Appendix A.

**Scale Aggregation.** We consider a general class of operations such as sum, max, and mean in our framework to aggregate the costs from multiple scales. Selecting an optimal aggregator will depend on the selected wavelet kernel, the selected set of scales, and the operation's discrimination abilities. The chosen operation is taken elementwise across all scales $S$.

**Remark 1** (*Relating WOT to Geodesic-Based Gromov-Wasserstein OT*). *Let the wavelet function be the heat kernel at a single scale $S = \{s\}$. For filters $\mathbf{F}^A = \mathbf{J}_n$ and $\mathbf{F}^B = \mathbf{J}_m$, in the limit $s \to 0^+$, the WOT distance reduces to the Gromov-Wasserstein distance with a geodesic-RBF kernel discrepancy. A proof is included in Appendix B. This highlights that under certain conditions, the Gromov-Wassertein OT framework is a subset of the WOT framework.*

We now propose two specific WOT implementations with particular choices for filters and optimization techniques.

## 3.3 ENTROPY-BASED WOT

E-WOT is an entropy-based heuristic for the filters $\mathbf{F}^A$ and $\mathbf{F}^B$. Intuitively, higher entropy scales may provide greater information and thereby should be emphasized by the filters. For each dataset $X \in \{A, B\}$ with corresponding wavelets $\psi^X$, the entropy is estimated at each scale $s$ using kernel density estimation (KDE). The entropy value for each scale is then employed as the respective filter for that scale:

$$\mathbf{F}^{(s,X)} = \mathcal{H}^{(s,X)}\mathbf{J}_{|X|} \quad \text{with} \quad \mathcal{H}^{(s,X)} = \mathbb{E}_i\left[-\ln \frac{1}{|X|}\sum_{j=1}^{|X|} K_h\left(\psi_{ij}^{(s,X)}\right)\right] \tag{5}$$

where $\mathcal{H}^{(s,X)} \in \mathbb{R}$ and $K_h$ is a kernel (not to be confused with the wavelet kernel) with smoothing parameter (or *bandwidth*) $h > 0$. In what follows, our attention is restricted to the Gaussian kernel for $K_h$.

We proceed to optimize E-WOT in the same spirit as Peyré et al. (2016) using projected gradient descent. If we augment our objective with an entropic regularization $H(\mathbf{T}) = -\sum_{i,j=1}^{n,m} \mathbf{T}_{ij}(\ln \mathbf{T}_{ij} - 1)$ (not to be confused with $\mathcal{H}^{(s,X)}$) with a weight $\varepsilon$, Proposition 2 in Peyré et al. (2016) has shown that the projection step reduces to solving the Sinkhorn distance (Cuturi, 2013). Therefore, we have

$$\text{E-WOT}(\psi^A, \psi^B, \mathbf{p}, \mathbf{q}, S) = \inf_{\mathbf{T} \in \prod(\mathbf{p},\mathbf{q})} \mathbf{C}(\psi^A, \psi^B, \mathbf{F}^A, \mathbf{F}^B, S, \mathbf{T}) - \varepsilon H(\mathbf{T})$$

with each projected gradient descent step as

$$\mathbf{T} \leftarrow \mathcal{T}\left(\text{agg}_{s \in S}\left[\mathbf{L}(\mathbf{F}^{(s,A)}\psi^{(s,A)}, \mathbf{F}^{(s,B)}\psi^{(s,B)}) \otimes \mathbf{T}\right], \varepsilon, \mathbf{p}, \mathbf{q}\right) \tag{6}$$

where $\mathcal{T}$ is a Sinkhorn projection. Note that E-WOT can be similarly defined in an *unbalanced* formulation (where there are different masses of $\mathbf{p}$ and $\mathbf{q}$) by replacing the Sinkhorn algorithm with the unbalanced counterpart proposed by Chizat et al. (2018). In Section 4 and Appendix D, we demonstrate the advantages of WOT over competing works in both balanced and unbalanced settings.

### 3.4 LEARNED WOT

Unlike the static and uniform filters of E-WOT, we introduce here an implementation of WOT called L-WOT such that the filters $\mathbf{F}^A$ and $\mathbf{F}^B$ are *learned*. In a minimization-maximization fashion, we alternate between minimizing the WOT objective with respect to the transport plan T and maximizing the WOT objective with respect to the filters $\mathbf{F}^A$ and $\mathbf{F}^B$. We hence augment the existing WOT objective, Equation (3), with an inner optimization step:

$$\text{L-WOT}(\psi^A, \psi^B, \mathbf{p}, \mathbf{q}, S) = \inf_{\text{T} \in \prod(\mathbf{p}, \mathbf{q})} \sup_{\mathbf{F}^A, \mathbf{F}^B} \text{C}(\psi^A, \psi^B, \mathbf{F}^A, \mathbf{F}^B, S, \text{T}) - \varepsilon H(\text{T}) \qquad (7)$$

$$\text{s.t. } ||\mathbf{F}^A - \mathbf{J}_n||_2 + ||\mathbf{F}^B - \mathbf{J}_m||_2 < \delta$$

Intuitively, filters that maximize the objective reveal portions of each space that the current transport plan does not match well, thus forcing the next transport optimization step to better match these regions. However, to restrict the optimization from finding trivial solutions (e.g. filters that noise the wavelets and scale the magnitude of the noise to $\infty$), an additional constraint is added that keeps the discrepancy between the filters and $\mathbf{J}$ below a threshold $\delta$. In practice, it is often beneficial to revise Equation (7) with $\mathbf{F}^X$ weighted by the squared root of the entropy $\tilde{\mathcal{H}}^X = \sqrt{\mathcal{H}^X}$.

L-WOT is optimized such that at each iteration, (1) we fix the filters and minimize $\text{C}(\cdot)$ with respect to the transport plan T using Sinkhorn iterations in the same way as Section 3.3 (note that the filter constraints do not need to be enforced here) and at another step (2) we fix T and maximize $\text{C}(\cdot)$ with respect to the filters $\mathbf{F}^A$ and $\mathbf{F}^B$ using gradient ascent. For step (2), we formulate the constrained objective in its dual form by taking the Lagrangian with multiplier $\lambda$. The optimization details are outlined in Algorithm 1.

---

**Algorithm 1** L-WOT Optimization

---

1: **Input:** SGWs $\psi^A$, SGWs $\psi^B$, marginal $\mathbf{p}$, marginal $\mathbf{q}$, scales $S$, BCD steps N, inner steps K
2: **Output:** Transport plan T
3: Initialize $\mathbf{F}^A = \mathbf{J}_n$, $\mathbf{F}^B = \mathbf{J}_m$, $\lambda_A$, $\lambda_B$, $\varepsilon$
4: **for** N loops **do**
5:     **for** K loops **do**
6:        $\tilde{\mathbf{F}}^A, \tilde{\mathbf{F}}^B = \tilde{\mathcal{H}}^A \circ \mathbf{F}^A, \tilde{\mathcal{H}}^B \circ \mathbf{F}^B$
7:        $\text{L} = \text{L}(\mathbf{F}^{(s,A)}\psi^{(s,A)}, \mathbf{F}^{(s,B)}\psi^{(s,B)})$
8:        Update $\text{T} = \mathcal{T}(\text{agg}_{s \in S}[\text{L} \otimes \text{T}], \varepsilon, \mathbf{p}, \mathbf{q})$
9:     **end for**
10:     **for** K loops **do**
11:        $\tilde{\mathbf{F}}^A, \tilde{\mathbf{F}}^B = \tilde{\mathcal{H}}^A \circ \mathbf{F}^A, \tilde{\mathcal{H}}^B \circ \mathbf{F}^B$
12:        $\text{Cost} = \text{C}(\psi^A, \psi^B, \mathbf{p}, \mathbf{q}, \tilde{\mathbf{F}}^A, \tilde{\mathbf{F}}^B, S, \text{T})$
13:        $\text{reg}_A = ||\mathbf{F}^A - \mathbf{J}_n||_2$, $\text{reg}_B = ||\mathbf{F}^B - \mathbf{J}_m||_2$
14:        $\mathbf{F}^A = \mathbf{F}^A + \nabla_{\mathbf{F}^A}(\text{Cost} - \lambda_A \text{reg}_A)$
15:        $\mathbf{F}^B = \mathbf{F}^B + \nabla_{\mathbf{F}^B}(\text{Cost} - \lambda_B \text{reg}_B)$
16:     **end for**
17: **end for**

---

## 4 EXPERIMENTS

With the primary goal of aligning unpaired single-cell data, we begin by evaluating the effectiveness of WOT in simpler cases that exhibit some challenges of single-cell data. WOT is first assessed on a point cloud matching experiment with increasing levels of noise and dropout. Afterward, we demonstrate WOT's ability to match points sampled from low-dimensional, highly non-isometric manifolds (i.e. animal shapes). We finally test WOT to align two real single-cell datasets with gene expression, chromatin accessibility, and DNA methylation profiles.

Note that since we are operating in a completely *unpaired* setting, we assume that we do not have access to a validation set. Hence, most hyperparameters are fixed to default values. However, for a small set of sensitive hyperparameters, we employ an unsupervised tuning procedure outlined in Algorithm 2. We provide the full set of relevant hyperparameters for WOT and the specific fixed values used within the experiments in Appendix C). We also have further guidance on hyperparameter selection including the wavelet filter $g$ and implementation of WOT (i.e. E-WOT versus L-WOT) in Appendix C.1

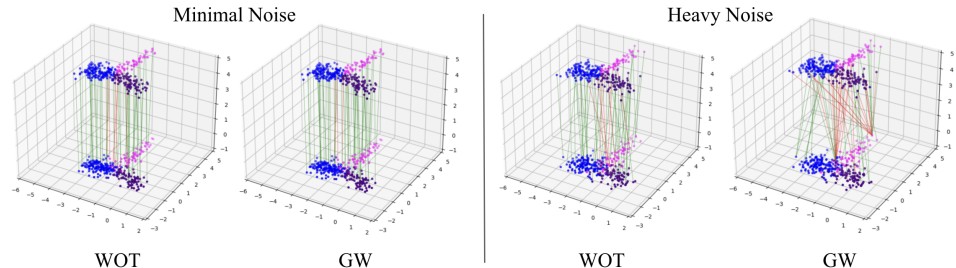

Figure 2: Matching two bifurcations of three classes (denoted by colors) with and without heavy noise. For each graph, we plot the first two principal components of each bifurcation dataset and add the z dimension to separate the two bifurcation datasets for illustration purposes. Green lines signify correct matches while red lines signify incorrect matches. (**Left**) with minimal noise, vanilla-WOT and GW perform similarly (**Right**) with heavy Gaussian noise (variance=0.1 of average pointwise distance), WOT still maintains high-quality matches while GW does not.

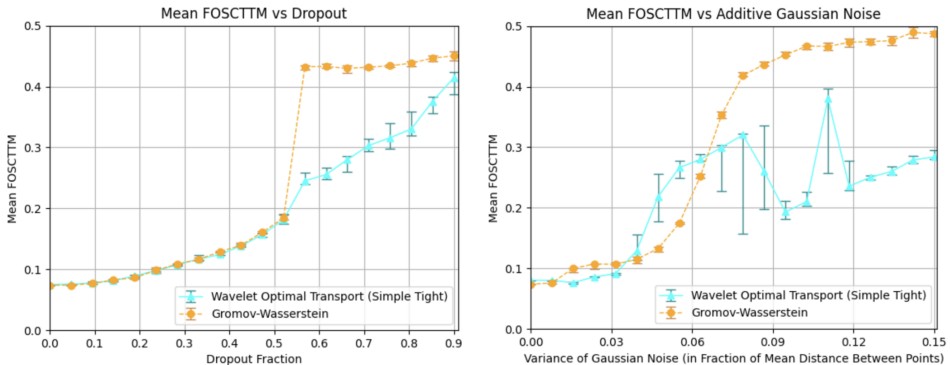

Figure 3: Comparing the robustness of vanilla-WOT and Gromov-Wasserstein OT to increasing levels of dropout and noise. Each dropout (**left**) level and additive noise (**right**) is performed ten times; the top, middle, and bottom of the error bars represent the 75th, 50th, and 25th percentile, respectively.

For all experiments, WOT uses the barycentric projection (Bonneel et al., 2016) based on the transport plan T to project points from one domain to another (results from other projection techniques are in Appendix D.2.1). Throughout all experiments, the discrepancy measure L is the squared loss $L(a, b) := \frac{1}{2}(a - b)^2$. Additionally, we primarily show results for the *balanced* datasets by computing E-WOT and L-WOT based on balanced Sinkhorn iterations. However, it is important to note that WOT can also handle *unbalanced* datasets; we provide additional results based on unbalanced Sinkhorn iterations in Appendix D. All experiments were conducted on one NVIDIA RTX A6000 machine.

## 4.1 BIFURCATION MATCHING

Cellular differentiation is a common biological process that single-cell instruments aim to capture. We start by demonstrating the effectiveness of WOT on a toy dataset from Liu et al. (2019) providing a bifurcation simulation that resembles the divergence of cell states. This dataset contains two sets $A$ and $B$ of $n = 300$ samples each. Each set aims to represent a distinct modality where $A$ has points with 1000 dimensions and $B$ has points with 2000 dimensions. Although the true pairing between points in $A$ and $B$ is known and used to assess the accuracy of OT methods, this information is not used when learning the transport plans.

For our first experiment, we add isotropic noise to $A$ and $B$ sampled from the Gaussian $\mathcal{N}(\mathbf{0}, \sigma^2 \mathbf{I}_{|X|})$. We compare the performance of WOT and GW-OT along various levels of noise $\sigma^2$ using the *fraction of samples closer than the true match* (FOSCTTM) (Liu et al., 2019). In prac-

| SHREC20 | TEST SET 1 | TEST SET 2 | TEST SET 3 | TEST SET 4 |
|---|---|---|---|---|
| E-WOT (HEAT KERNEL) | 0.786/0.172 | 0.551/0.422 | 0.533/0.294 | **0.639**/0.253 |
| E-WOT (SIMPLE TIGHT) | 0.669/0.215 | 0.542/0.424 | **0.575/0.268** | 0.578/0.282 |
| E-WOT (MEYER) | 0.641/0.219 | 0.530/0.431 | **0.575**/0.270 | 0.582/0.287 |
| L-WOT (HEAT KERNEL) | **0.790/0.171** | **0.729/0.209** | 0.500/0.388 | 0.623/0.264 |
| SCOT (GW) | 0.710/0.208 | 0.335/0.600 | 0.458/0.298 | 0.635/**0.251** |
| SCOTv2 (UNBALANCED GW) | 0.572/0.258 | 0.631/0.375 | 0.479/0.311 | 0.545/0.272 |
| UNIONCOM | 0.536/0.274 | 0.127/0.787 | 0.329/0.416 | 0.553/0.281 |
| PAMONA | 0.383/0.362 | 0.110/1.384 | 0.204/0.665 | 0.221/0.440 |

Table 1: Relative Geodesic Error on SHREC20 dataset reported as (% matches $< 0.25 \uparrow$) / (Mean $\downarrow$). Each test set holds shapes of decreasing isometry from test set 1 with the *highest* isometry to test set 4 with the *lowest* isometry. The best performing method for each test set is **bolded**.

tice, we add noise to each point relative to the average distance between samples in each dataset: $\sigma^2 \in [0.0, 0.15] \times$ (avg dist). As shown in the bottom graph of Figure 3, WOT and GW-OT perform similarly in low noise levels $(0.00 - 0.065)$, but WOT maintains significantly better performance in medium and high noise levels $(0.065 - 0.15)$. The same trend is clear in another experiment (top graph of Figure 3) where we introduce *dropout* in the bifurcation dataset and evaluate the methods' ability to find accurate matches. We revise the conventional definitions of dropout to a more difficult scenario: rather than removing a point entirely, we instead add a large amount of noise to that point such that it loses its meaning *and* muddles the rest of the dataset. Specifically, we randomly select a fraction of samples where noise is added with a variance that is equal to the average distance between samples while keeping the unselected fraction of samples the same. This dropout is applied independently in *both* datasets $A$ and $B$. Similarly to the additive noise experiment, GW-OT and WOT perform approximately the same in lower regimes of dropout, while WOT outperforms GW-OT in higher regimes of dropout.

Although WOT achieves better performance overall, it is important to note that the variance of WOT's mean FOSCTTM in the additive noise experiment is significantly greater than GW; this result could imply that our method may require further hyperparameter tuning. Interestingly, we do not see this high variance in WOT's performance in the dropout experiment.

### 4.2 SHAPE CORRESPONDENCE

Cell states are believed to lie on a low-dimensional manifold (Moon et al., 2018) in the data space. However, each data modality (i.e. single-cell profiling technology) carries its own variabilities such as the color of nuclei images or the read depth of scRNA-seq that may respectively distort the common manifold that the cells share. In other words, the structures of each modality's data manifold will have similarities yet be highly different. We analogize this with the problem of point matching of highly non-isometric shapes (low-dimensional manifolds). SHREC20 (Dyke et al., 2020) provides four sets of increasingly different pairs of animals with ground truth correspondences between key landmarks on each shape (e.g. ears, tails, and legs).

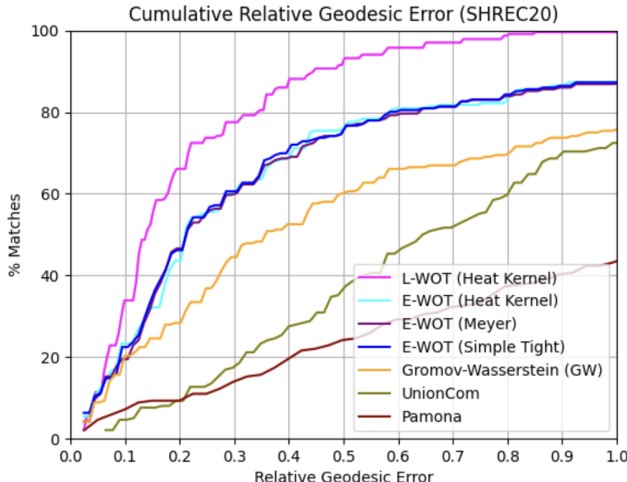

Figure 4: Quality of correspondences on SHREC20's test set 2 as measured by cumulative relative geodesic error. We plot the percentage of projected samples from one animal to another that are within $x$ relative geodesic distance from the ground truth.

For each animal, we sample 1000 points and combine these points with the ground truth points as input to the evaluation methods. We then calculate the relative geodesic error $\epsilon(\mathbf{a}_i) = d_Y(\mathbf{a}_i, \mathbf{b}_i) / \text{area}(B)^{\frac{1}{2}}$ of projecting $\mathbf{a}_i$ from animal $A$ onto animal $B$ compared with the ground truth point $\mathbf{b}_i$ on animal $B$.

Table 1 records the mean relative geodesic error and the percent of matches that are less than 0.25 relative geodesic error of WOT and competing works. Particularly for test set 2, WOT provides a very large improvement in performance compared to previous methods, as shown in Figure 4. Additionally, L-WOT obtains substantially better quality of correspondences than E-WOT while the performance between different filters for E-WOT remains approximately the same. This difference between L-WOT and E-WOT highlights that applications may find it more useful to leverage one particular implementation of WOT. We also see that WOT and GW-OT roughly have a logarithmic pattern in Figure 4; the sharper "elbow" of WOT demonstrates that WOT is able to achieve correspondences with a smaller error margin more quickly than GW-OT. As the relative geodesic error increases, the curves of both methods tend to level off. However, our method maintains a consistent lead, indicating that even as the error tolerance increases, WOT continues to match more samples within the error threshold.

The performance comparisons on the other test sets are included in the Appendix D. We omit comparisons with shape-specific matching methods since they cannot be scaled to higher dimensions than 3D and would therefore not be useful for single-cell modality alignment.

### 4.3 ALIGNING SINGLE CELL DATASETS

We now evaluate WOT on two real single-cell multi-omic datasets, scGEM and SNARE-seq. scGEM charts the trajectory of human somatic cells during reprogramming to induced pluripotent stem cells (iPSCs); the data is produced using the scGEM co-assay, concurrently capturing both the scRNA-seq and DNA methylation profiles of the cells. SNARE-seq is collected from human fibroblast cells (BJ), human embryonic cells (H1), human erythroleukemia cells (K562), and human lymphoblastoid cells (GM12878) using the SNARE-seq co-assay, profiling both scRNA-seq and chromatin accessibility simultaneously. Unlike scGEM which is undergoing cellular reprogramming, the SNARE-seq dataset exhibits more distinct clusters between the different cell types.

| LABEL TRANSFER ACCURACY | SNARE-SEQ | scGEM |
|---|---|---|
| E-WOT (HEAT KERNEL) | **0.961** | 0.472 |
| E-WOT (SIMPLE TIGHT) | 0.881 | 0.492 |
| L-WOT (HEAT KERNEL) | 0.774 | 0.528 |
| L-WOT (SIMPLE TIGHT) | 0.803 | **0.616** |
| SCOT | 0.852 | 0.423 |
| SCOTv2 | 0.826 | 0.509 |
| UNIONCOM | 0.411 | 0.332 |
| PAMONA | 0.554 | 0.385 |
| MMD-MA | 0.523 | 0.360 |
| PAMONA | 0.554 | 0.385 |
| CROSS AE | 0.511 | 0.363 |
| BINDSC | 0.713 | 0.387 |
| SEURAT | 0.423 | 0.408 |

Table 2: Label transfer accuracy of WOT and competing methods on single-cell multi-omic datasets. We use the results reported by Demetci et al. (2022a) for competing methods. The best performing method is **bolded**.

We follow the same preprocessing steps as Demetci et al. (2022b) for the two datasets and use label transfer accuracy (Cao et al., 2020) as our evaluation metric. The resulting scGEM dataset after preprocessing has 177 samples with 34 dimensions for the gene expression data and 27 dimensions for the methylation data. For SNARE-seq, the resulting dataset has 1047 samples with 19 dimensions for the chromatin accessibility data and 10 dimensions for the gene expression data.

Table 2 reflects a significant improvement in accuracy by WOT in both datasets. We also observe that while there is significant variability in the performance between E-WOT and L-WOT as well as the specific wavelet kernel used, they all exceed or are on par with the current state-of-the-art methods. Additionally, L-WOT performs much better than E-WOT and existing methods on the scGEM while the inverse is seen in SNARE-seq. A potential reason for this difference is that scGEM profiles cells in dedifferentiation, so the boundaries of cell types are not as clear as those of SNARE-seq where cell

type clusters are distinct—this could imply that L-WOT is better equipped for single-cell datasets with experiment setups like scGEM while E-WOT is better for single-cell datasets like SNARE-seq.

## 5 CONCLUSION

We presented Wavelet Optimal Transport, a framework that leverages spectral graph wavelets to better align unpaired datasets. Through initial experiments, we demonstrated that WOT is able to maintain high-quality matches across datasets even in the presence of high noise, dropout, and non-isometry. Finally, we showed the effectiveness of WOT on two real single-cell datasets, outperforming the previously most accurate methods.

A fruitful direction for future research is to incorporate the WOT objective in machine learning pipelines as a loss function such that we can match data distributions at specific bands of scales or space. WOT could also be applied in linking cross-section time series data where cross-sections may be collected with different modalities; this problem is common in trajectory analysis or modeling perturbation response in cells. Another direction that warrants more investigation is the design of more effective filters that could be used in our framework.

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

## A   SPECTRAL GRAPH WAVELET CONSTRUCTION

For a dataset $X = \{\mathbf{x}_i\}_{i=1}^n$, we compute the spectral graph wavelets (SGWs) as follows: (1) build a fully connected weighted adjacency matrix $W$, (2) calculate the normalized graph Laplacian $\mathcal{L}$, and (3) compute the spectral graph wavelets of $\mathcal{L}$ using Chebyshev's polynomial approximation (Hammond et al., 2011) implemented in Python by PyGSP (Defferrard et al.) .

### A.1   FULLY CONNECTED WEIGHTED ADJACENCY MATRIX

We construct $W$ with weights between nodes $i$ and $j$ given by the RBF-kernel:

$$W_{ij} = \mathrm{RBF}(\mathbf{x}_i, \mathbf{x}_j) = \exp\left(-\frac{\tilde{d}_X(\mathbf{x}_i, \mathbf{x}_j)^2}{2\sigma^2}\right) \tag{8}$$

where $\tilde{d}_X$ is chosen to be the approximate geodesic distance calculated by finding the shortest path between pairs of nodes on the $k$NN graph of $X$ (where $k$ is a hyperparameter given in Appendix C). The value of $\sigma$ is set as the median of this distance between all pairs of points in $X$.

## B   PROOFS

**Remark 1** (*Relating GW using Geodesic Distance and WOT*). *Let the wavelet function be the heat kernel at a single scale $S = \{s\}$. For filters $\mathbf{F}^A = \mathbf{J}_n$ and $\mathbf{F}^B = \mathbf{J}_m$, in the limit $s \to 0^+$, the WOT distance reduces to the Gromov-Wasserstein distance with a geodesic-RBF kernel discrepancy.*

**Proof.** Recall the GW discrepancy as

$$\mathrm{GW}(\mathbf{p}, \mathbf{q}) = \inf_{\mathrm{T} \in \Pi(\mathbf{p}, \mathbf{q})} \sum_{i,k=1}^n \sum_{j,l=1}^m \mathrm{L}(d_A(\mathbf{a}_i, \mathbf{a}_k), d_B(\mathbf{b}_j, \mathbf{b}_l))\mathrm{T}_{ij}\mathrm{T}_{kl} . \tag{9}$$

where we have two metric spaces $(A, d_A)$ and $(B, d_B)$ with probability measures $\mathbf{p}$ and $\mathbf{q}$, respectively. $d_A$ and $d_B$ are the respective geodesic distances between the points in $A$ and the points in $B$. Now, since $\mathbf{F}^{s,A} = \mathbf{J}_n$ and $\mathbf{F}^{s,B} = \mathbf{J}_m$ at a single scale $S = \{s\}$, WOT reduces to

$$\mathrm{WOT} = \inf_{\mathrm{T} \in \Pi(\mathbf{p}, \mathbf{q})} \sum_{i,k=1}^n \sum_{j,l=1}^m \mathrm{L}(\psi_{ik}^{(s,A)}, \psi_{jl}^{(s,B)})\mathrm{T}_{ij}\mathrm{T}_{kl} \tag{10}$$

where $\psi$ is the heat kernel. Recall Varadhan's Lemma which relates the heat kernel and geodesic distance as $-4s\log(\psi_{x,y}^{(s)}) \simeq d_X^2(x, y)$ for $t \simeq 0^+$. We can rewrite the heat kernel with respect to the squared geodesic distance as

$$\psi_{xy}^{(s)} \simeq \exp(-\frac{d_X^2(x, y)}{4t}) . \tag{11}$$

Thus, we have the WOT formulation in terms of the geodesic distance as

$$\mathrm{WOT} \simeq \inf_{\mathrm{T} \in \Pi(\mathbf{p}, \mathbf{q})} \sum_{i,k=1}^n \sum_{j,l=1}^m \mathrm{L}(\exp(-\frac{d_A^2(\mathbf{a}_i, \mathbf{a}_k)}{4t}), \exp(-\frac{d_B^2(\mathbf{b}_j, \mathbf{b}_l)}{4t}))\mathrm{T}_{ij}\mathrm{T}_{kl} \tag{12}$$

which is equivalent to the GW discrepancy if the metric for space $A$ is redefined as the RBF kernel $:= \exp(-\frac{d_A^2(\mathbf{a}_i, \mathbf{a}_k))}{4t})$ and space $B$ is redefined as the RBF kernel $:= \exp(-\frac{d_B^2(\mathbf{b}_j, \mathbf{b}_l)}{4t})$. Proof completed.

## C   HYPERPARAMETERS

We provide the set of hyperparameter values used in each experiment. Note that most values were fixed during experiments since we did not have validation data to conduct hyperparameter tuning.

---

**Algorithm 2** Unsupervised Hyperparameter Selection Procedure

---

**Require:** Source and target datasets $X, Y$
**Ensure:** Hyperparameters $\varepsilon$, agg, norm
   $\varepsilon \leftarrow 10^{-4}$, agg $\leftarrow$ sum, norm $\leftarrow$ RBF, $\eta \leftarrow 10^{-6}$
  **while** true **do**
    $T \leftarrow$ COMPUTETRANSPORT$(X, Y, \varepsilon, $ agg, norm$)$
    $U_{ij} \leftarrow \frac{1}{mn}, \forall i, j$ where $m, n$ are dimensions of T
    **if** $\neg\exists_{i,j}$isNaN$(T_{ij}) \wedge \|T - U\|_F > \eta$ **then**
      **return** $\varepsilon$, agg, norm
    **end if**
    **if** $\varepsilon < 1.0$ **then**
      $\varepsilon \leftarrow \varepsilon + 0.5 \cdot 10^{\lfloor \log_{10}(\varepsilon) \rfloor}$
    **else if** agg = sum **then**
      agg $\leftarrow$ mean, $\varepsilon \leftarrow 10^{-4}$
    **else if** agg = mean **then**
      agg $\leftarrow$ max, $\varepsilon \leftarrow 10^{-4}$
    **else if** norm = RBF **then**
      norm $\leftarrow$ L2, $\varepsilon \leftarrow 10^{-4}$
    **else**
      **raise** Error
    **end if**
  **end while**

---

However, in some instances, improper hyperparameters can lead to an invalid uniform or NaN transport plan. In such cases, we adjust the entropic regularization parameter $\varepsilon$, aggregation scheme agg, and the weight normalization according to Algorithm 2

| ALL EXPERIMENTS | HYPERPARAMETER VALUES |
|---|---|
| $\varepsilon$ REGULARIZATION (PEYRÉ ET AL., 2016) | 0.001 |
| AGGREGATION OPERATION | SUM |
| WEIGHT NORMALIZATION | RBF |
| WOT OUTER ITERATIONS (N) | 100 |
| SINKHORN INNER ITERATIONS (K) | 100 |
| GAUSSIAN KDE BANDWIDTH (H) | 0.4 |
| NUMBER OF SCALES | 20 |
| $\rho_1$ (UNBALANCED) | 1.0 |
| $\rho_2$ (UNBALANCED) | 1.0 |
| LANGRANGE MULTIPLER $\lambda_A, \lambda_B$ | 2.0 |

Table 3: Default hyperparameters for WOT.

| Hyperparameter | Specification |
|---|---|
| $kNN$ | Fixed from Demetci et al. (2022b) Fig. S4 |
| $\sigma_{\text{RBF}}$ | median($\{d_{ij}\}$) heuristic |
| Number of Scales | Fixed from prelim. development |
| $\varepsilon$ | Algorithm 2 determined |
| agg | Algorithm 2 determined |
| $g(x)$ | Experiment 1: defaulted to simplest wavelet; Experiment 2,3: multiple wavelets tested |
| Wavelet hyperparameters | PyGSP defaults Defferrard et al. |
| $\delta_{\text{L-WOT}}$ | Fixed from prelim. development |
| $h_{\text{KDE}}$ | Fixed from prelim. development |

Table 4: Hyperparameter Selection Process

| TOY DATASET | HYPERPARAMETER VALUES |
|---|---|
| $\varepsilon$ REGULARIZATION (NOISE) | 0.001 |
| $\varepsilon$ REGULARIZATION (DROPOUT) | $0.0005(0.0 - 0.5\%), 0.0001(0.6 - 0.9\%)$ |
| AGGREGATION OPERATION | MEAN |
| WAVELET KERNEL | SIMPLE TIGHT |
| METRIC | EUCLIDEAN DISTANCE |
| WEIGHT NORMALIZATION | RBF |

| SHREC20 DATASET | HYPERPARAMETER VALUES |
|---|---|
| $\varepsilon$ REGULARIZATION | 0.1 |
| AGGREGATION OPERATION | SUM |
| WAVELET KERNEL | MULTIPLE |
| METRIC | APPROXIMATE GEODESIC |
| $k$ IN $kNN$ FOR GEODESIC | 30 |
| UNBALANCED $\rho$ (SÉJOURNÉ ET AL., 2021) | 1.0 |
| WEIGHT NORMALIZATION | RBF |

| SNARE-SEQ | HYPERPARAMETER VALUES |
|---|---|
| $\varepsilon$ REGULARIZATION | 0.01 (HEAT-EWOT) |
| | 0.1 (SIMPLE TIGHT-LWOT), 0.05 (ELSE) |
| AGGREGATION OPERATION | SUM |
| WAVELET KERNEL | MULTIPLE |
| METRIC | APPROXIMATE GEODESIC |
| $k$ IN $kNN$ FOR GEODESIC | 30 |
| WEIGHT NORMALIZATION | L2 |

| SCGEM | HYPERPARAMETER VALUES |
|---|---|
| $\varepsilon$ REGULARIZATION | 0.05 (SIMPLE TIGHT-EWOT), 0.01 (ELSE) |
| AGGREGATION OPERATION | SUM |
| WAVELET KERNEL | MULTIPLE |
| METRIC | APPROXIMATE GEODESIC |
| $k$ IN $kNN$ FOR GEODESIC | 30 |
| WEIGHT NORMALIZATION | RBF |

Table 5: Hyperparameter values in all reported experiments. While experiments using unbalanced Sinkhorn are not reported in the main paper, and therefore do not utilize the unbalanced hyperparameter $\rho$, additional experiments were conducted on SHREC20 using unbalanced Sinkhorn, whose results are reported in Appendix D. For the toy dataset, recall that we use vanilla-WOT where the filters $\mathbf{F} = \mathbf{J}$. Also, note that $\lambda$ hyperparameter value is only relevant in the experiments evaluating L-WOT, and thus not all the experiments include that hyperparameter. "Multiple" refers to evaluating multiple wavelet kernels for a given experiment. Lastly, "BOTH" refers to both L-WOT and E-WOT.

### C.1 GUIDANCE ON CHOOSING HYPERPARAMETERS

In practice, most of the hyperparameters in WOT can be fixed to default values, leaving only two key components that may require more careful consideration:

1. **The choice between L-WOT and E-WOT implementations**: This decision can be made based on the characteristics of the dataset and the desired balance between adaptivity and computational efficiency. In many cases, users can start with a default implementation (e.g., E-WOT) and explore the alternative if the results are not satisfactory.

2. **The choice of wavelet kernel** $g$: In many practical scenarios, users can rely on prior knowledge or default choices like the heat kernel. Our experiments have shown that WOT consistently improves performance over existing methods, regardless of the specific kernel choice.

Additionally, we only primarily used the sum aggregation scheme within the WOT framework and have not observed other aggregation schemes being more effective. All other hyperparameters, such as the entropic regularization parameter, are common to any GW method. These hyperparameters can be selected using our proposed heuristic or other established heuristics in the literature, such as those presented in (Demetci et al., 2022b).

### C.2 HYPERPARAMETER TUNING FOR BASELINES

**Experiment 1.** For Gromov-Wasserstein, we adjust $\epsilon$ using the same strategy provided in Algorithm 2. All other relevant hyperparameters match Table 3.

**Experiment 2.** For Gromov-Wasserstein, UnionCom, and Pamona, they all share $\epsilon$ as a common hyperparmater. Thus, we adjust $\epsilon$ using Algorithm 2 but fix all other hyperparameters to their default values provided by the methods.

**Experiment 3.** Baseline results are taken from Demetci et al. (2022a), so we refer readers to this work for further details on hyperparameter selection.

## D ADDITIONAL EXPERIMENTAL RESULTS & FIGURES

### D.1 SHREC20 SHAPE CORRESPONDENCE

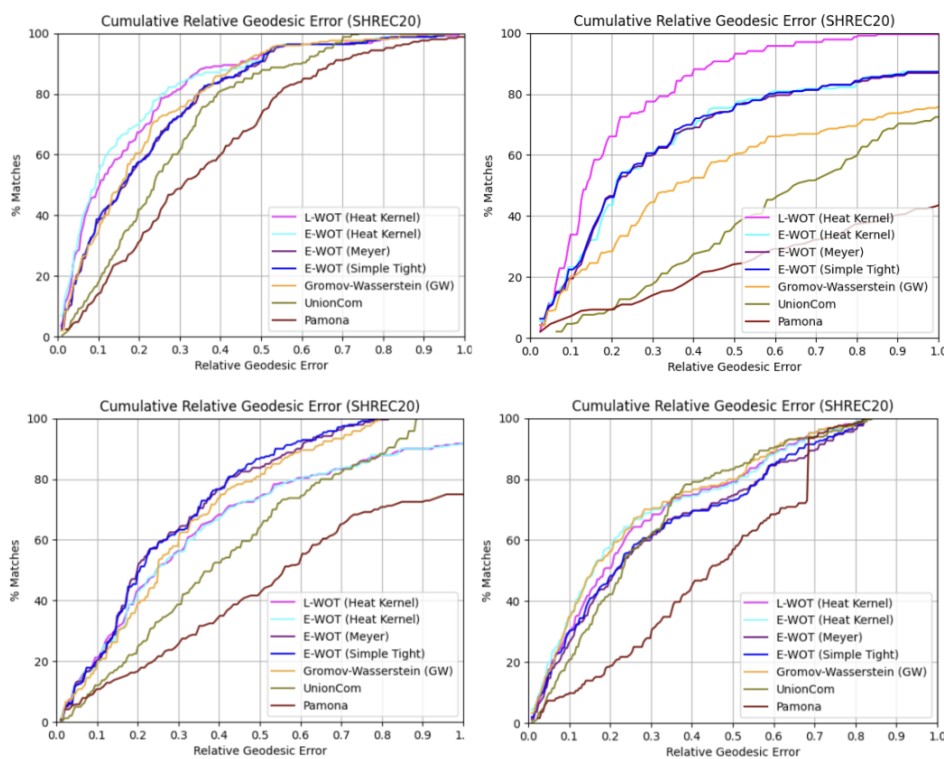

Figure 5: Cumulative Relative Geodesic Error of Correspondences on SHREC20's four test sets of increasing non-isometry using *balanced* formulation (**top left**) test set 1, lowest non-isometry (**top right**) test set 2, low non-isometry (**bottom left**) test set 3, high non-isometry (**bottom right**) test set 4, highest non-isometry.

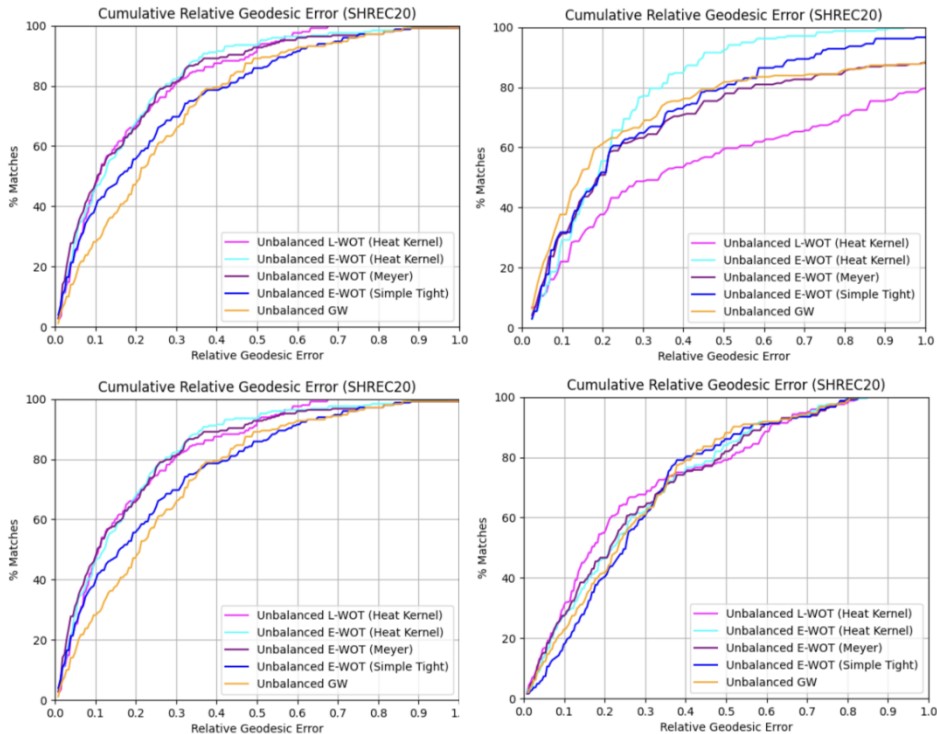

Figure 6: Cumulative Relative Geodesic Error of Correspondences on SHREC20's four test sets of increasing non-isometry using *unbalanced* formulation (**top left**) test set 1, lowest non-isometry (**top right**) test set 2, low non-isometry (**bottom left**) test set 3, high non-isometry (**bottom right**) test set 4, highest non-isometry.

## D.2 SCGEM & SNARE-SEQ

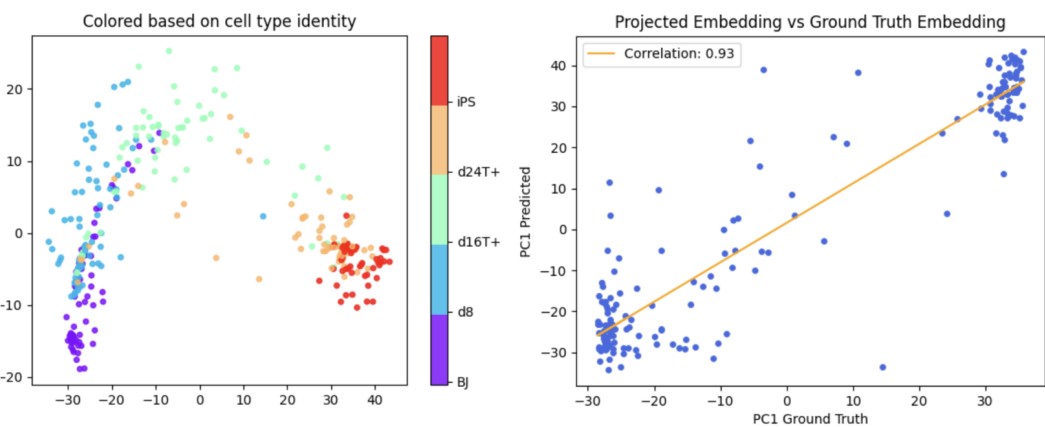

Figure 7: scGEM dataset alignment visualizations (**left**) we project gene profiling data into the DNA methylation data space and plot the first two principal components with their corresponding cell identity (**right**) after projection, we plot the first principal component of the ground truth point versus the first principal component of the corresponding projected point.

Additionally, for Section 4.3, we calculate FOSCTTM (introduced by (Liu et al., 2019)), which is a measure of the alignment error between two datasets. It quantifies the proportion of samples in one

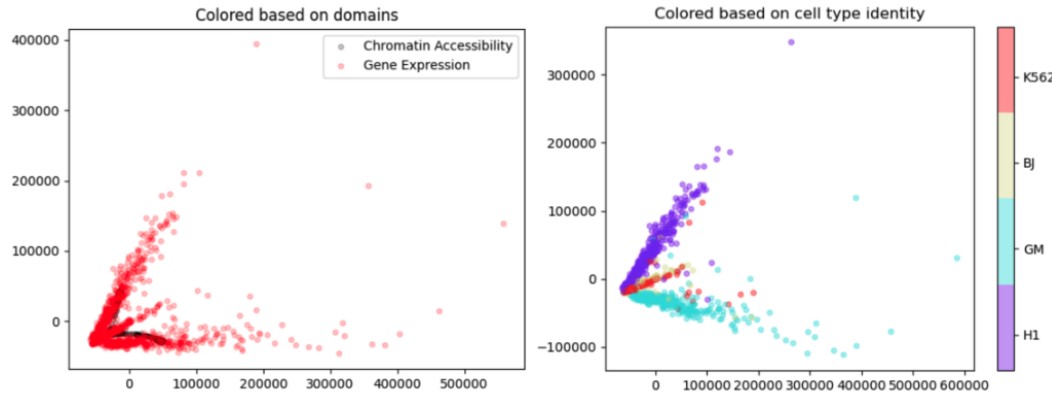

Figure 8: SNARE-seq dataset alignment visualizations where gene profiling data is projected into the ATAC-seq data space and plot the first two principal components with their corresponding cell identity.

dataset that are closer to a given sample in the other dataset than its true match, averaged across all samples in both datasets. The results are shown in the table below:

| FOSCTTM | scGEM | SNARE-seq |
|---|---|---|
| E-WOT (heat kernel) | 0.197 | 0.216 |
| E-WOT (simple tight) | 0.210 | 0.243 |
| L-WOT (heat kernel) | 0.202 | 0.262 |
| L-WOT (simple ticht) | 0.217 | 0.272 |
| SCOT (Demetci et al., 2022b) | 0.209 | 0.218 |
| MMD-MA | 0.437 | 0.473 |
| UnionCom | 0.691 | 0.510 |

Note that the results for the baseline methods are taken directly from (Demetci et al., 2022b). To ensure a fair comparison, we have followed the same hyperparameter settings that were used to obtain the results in Table 2 of our manuscript when computing LTA for WOT. It is worth noting that FOSCTTM is just one of the evaluation metrics, and our primary focus has been on label transfer accuracy (LTA) as reported in Table 2 of our manuscript since it is more representative of the metrics that are used in true unpaired alignment.

### D.2.1 ALTERNATIVE PROJECTION TECHNIQUES

We conducted additional experiments on the SNARE-seq and scGEM datasets where we replace the barycentric projection with the shared embedding projection approach proposed by (Cao et al., 2020). The results of these experiments are as follows:

| Label Transfer Accuracy | SNARE | ScGEM |
|---|---|---|
| E-WOT (Heat Kernel) w/ Shared Embedding (Cao et al., 2020) | 0.942 | 0.565 |
| E-WOT (Simple Tight) w/ Shared Embedding (Cao et al., 2020) | 0.941 | 0.616 |
| L-WOT (Heat Kernel) w/ Shared Embedding (Cao et al., 2020) | 0.939 | 0.627 |
| L-WOT (Simple Tight) w/ Shared Embedding (Cao et al., 2020) | 0.916 | 0.706 |

Comparing the label transfer accuracy (LTA) values in the above table to those reported in Table 2 of our manuscript, we observe a significant increase in LTA across both datasets and across the WOT implementations when using the shared embedding projection. For instance, on the SNARE-seq dataset, we see consistent LTA values of above 0.9 with the shared embedding projection, compared

to only a single implementation (E-WOT using heat kernel) achieving above 0.9 LTA with barycentric projection. On the scGEM dataset, L-WOT (Simple Tight) attains an LTA of 0.706 with the shared embedding projection, surpassing the 0.616 LTA obtained with barycentric projection.

The improvement in LTA suggests that while obtaining an informative transport plan is crucial for accurate alignment, the projection technique used to map the samples between the datasets also plays an important role. It is possible that even with an optimal transport plan, there may be an upper limit to the alignment quality achievable without an equally effective projection method.

### D.2.2 Analyzing Wavelet Scales and Filter Weights

To better understand why and when different implementations of WOT (EWOT vs LWOT) perform better, we empirically analyze (1) the wavelets corresponding to each scale of the single-cell datasets and (2) the weights of the filters in EWOT and LWOT and its impact on the wavelets.

**Wavelet Scales**. For each single-cell dataset and modality, we separate the spectral graph wavelets into their specific scale ranging from 1 to 20. Intuitively, larger valued scales (i.e. 20) represent high-frequency or local information while smaller valued scales (i.e. 1) represent low-frequency or global information. Since we only have the pairwise affinity matrix provided by the wavelets, we take the inverse and apply multidimensional scaling (MDS) in two dimensions. The resulting plots are shown below:

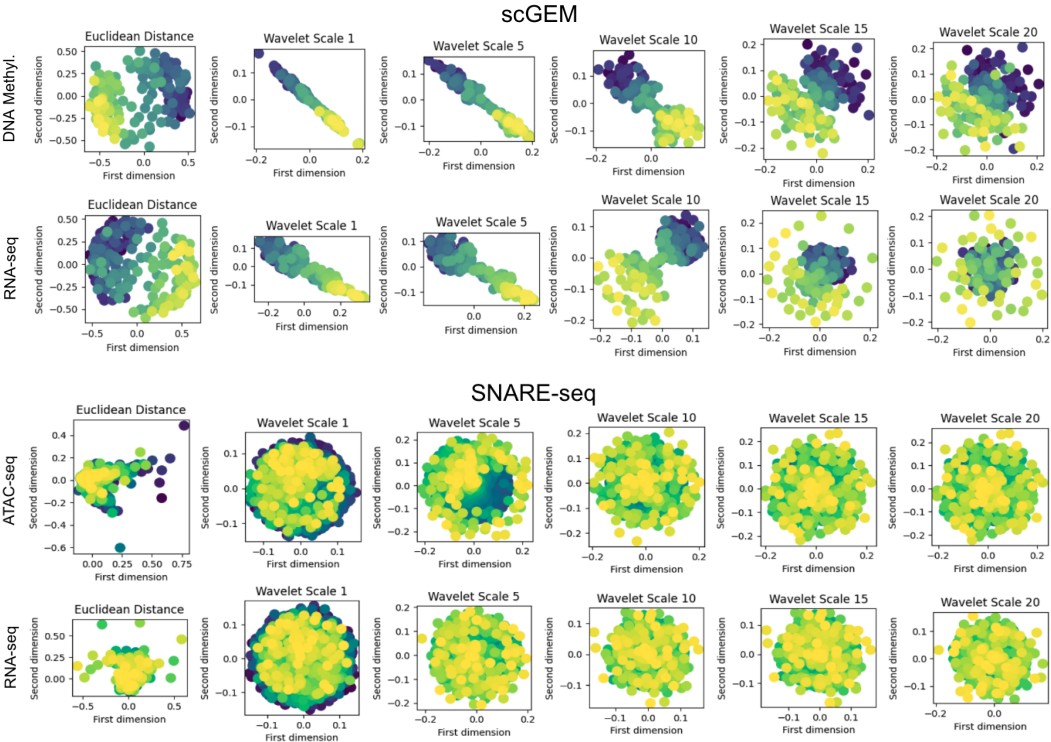

The color of each point represents the paired samples between each dataset. Ideally, we would want points of the same color to be in the same position in different plots. On the leftmost column, we visualize the 2D MDS embeddings of the original datasets where the pairwise distance matrix is given by Euclidean distance. Then, for every column to the right, we get progressively larger in scale value (i.e. the more right columns represent higher frequency scales). For instance, in scGEM, it is clear that the smaller-scale wavelets better reveal the samples that should be aligned while the larger-scaled wavelets are noised together.

Likewise, for SNARE-seq, the smaller scales better reveal the samples that should be paired while the larger scales muddle all the points, making alignment more ambiguous. Ideally, the filters should remove the wavelet scales that muddle the alignment while emphasizing the scales that provide a coherent structure for easier alignment.

**Filter Scales**. Since filters control which wavelet scales are used to align the datasets, it is necessary to interrogate which scales are filtered away or emphasized. We begin by plotting the distribution of filter values with respect to scales. For each scale, we take the maximum filter value. With the ground truth pairings, we learn an *ideal* filter (aka given this filter in WOT, we would have completely accurate alignment) that we use to compare with EWOT and LWOT.

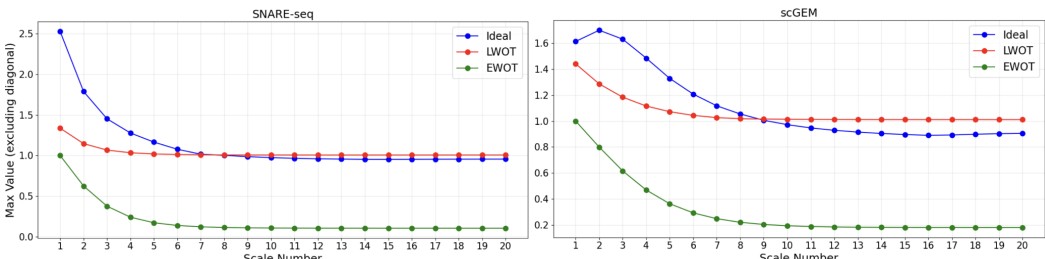

As shown in the figure above, the ideal filter has higher values concentrated at lower scale values for both datasets, which means that lower-valued wavelet scales are more important than higher-valued wavelet scales for perfect alignment. This emphasis on lower-valued wavelet scales makes sense based on our observations in the previous "Wavelet Scales" section where we established that lower-valued scales have more informative structures for accurate alignment.

Both E-WOT and L-WOT reflect similar trends of emphasizing lower-valued wavelet scales, explaining the performance improvement compared to baseline methods in Section 4.3.

We further explore the impact of filters on wavelets and ultimate alignment by visualizing the aggregated wavelets of each modality *after* the filters have been applied. In contrast to the previous section ("Wavelet Scales") which visualized each unfiltered wavelet scale individually, the below figures are both filtered and summed according to Equation (3). Each figure shows the 2D MDS embeddings of the filtered and aggregated pairwise distance matrix given by the inverse wavelet matrices.

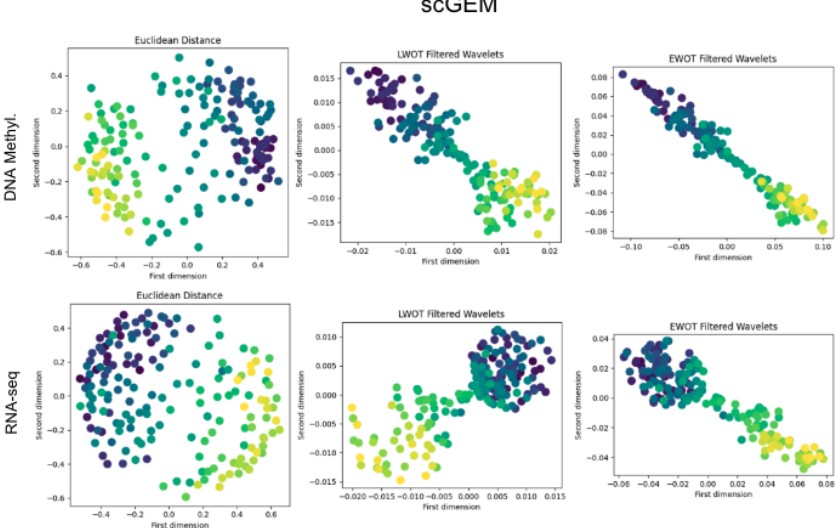

Since both EWOT and LWOT have been shown to emphasize lower-valued wavelet scales in the scGEM, it is unsurprising that the aggregated and filtered wavelets have similar structure to the lower-valued wavelet scales in the previous section ("Wavelet Scales"). Compared to the Euclidean case, the separation of points for the filtered wavelets (in both EWOT and LWOT) that correspond to each other is much clearer (e.g. points at one end of the DNA methylation modality corresponds to the same point at the end of the RNA-seq modality).

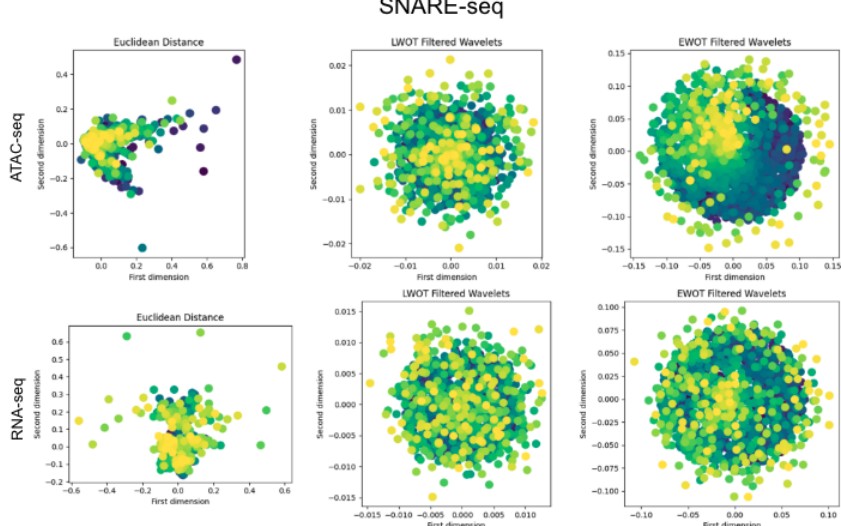

For SNARE-seq, the separation is not as clear as scGEM, but we still see that the filtered and aggregated wavelets have similar patterns to the lower-valued wavelet scales in the previous section.

While it is not clear how to quantify or predict when EWOT or LWOT would perform better, we now have intuition on why they perform better in specific datasets: the filters obtained by the implementation more closely match the *ideal* filters, which would provide better alignment. From this analysis, we can also observe why WOT performs better than GW and baselines that do not leverage multiple scales and filters: different scales of the dataset better reflect the geometric structure for accurate alignment while filters prune the scales that muddle the geometric structure. Baselines like GW only view the dataset at a single scale, disregarding the significant scale-specific geometric structures.

## E  TIME COMPLEXITY ANALYSIS

The runtime of our method is dominated by two steps: (1) the construction of the spectral graph wavelets and (2) optimizing E-WOT or L-WOT. We provide further analysis of WOT's complexity and how our method scales with respect to different input databases and parameterizations of WOT.

As is common with optimal transport frameworks, the efficiency of WOT can diminish as the volume of the dataset increases. Gromov-Wasserstein OT with entropic regularization (Peyré et al., 2016) scales $\mathcal{O}(n^3)$ with $n$ as the number of samples. Since our objective for vanilla-WOT and E-WOT is optimized similarly to GW-OT, we inherit the cubic complexity with an additional factor of $|S|$ because we recalculate Proposition 1 in (Peyré et al., 2016) $s \in S$ times, resulting in $\mathcal{O}(|S|n^3)$ time complexity. We further need to consider the $\mathcal{O}(n^2 + |S|n)$ computational complexity of calculating $\psi$ for $S$ scales using Cheyshev's polynomial approximation (Hammond et al., 2011). L-WOT would inherent greater complexity due to the nested loop with an inner subroutine that requires running E-WOT and automatic differentiation.

Specifically, we have

| Runtime | Scaling w.r.t feature dim. | Scaling w.r.t # of samples | Scaling w.r.t # of scales | Scaling w.r.t choices of wavelet kernels |
|---|---|---|---|---|
| Wavelet Construction | O($n^2 + |S|n$) | Constant | Quadratic | Linear |
| E-WOT | O($|S|n^3$) | Constant | Cubic | Linear |
| L-WOT | O($|S|n^3m$) | Constant | Polynomial | Linear |
| GW-OT (Peyré et al., 2016) | O($n^3$) | Constant | Cubic | N/A |

where $m$ is the complexity of running E-WOT and automatic differentiation.

- Wavelet Construction: Since the spectral graph wavelets leverage the pairwise distances of samples, constructing the spectral graph wavelets scales quadratically with the number of samples, but does not scale with feature dimensions (pairwise distances are a preprocessing step). When $n$ is large, the effects of the number of samples dominate the effects of the number of scales, so the overall runtime of constructing the spectral graph wavelets scales comparably to GW-OT.

- E-WOT: In practice, we often fix the number of scales used to construct the spectral graph wavelets to a small constant ($|S| = 20$), so the overall runtime of E-WOT scales comparably to GW-OT.

- L-WOT: The runtime of constructing wavelets becomes negligible with E-WOT

- L-WOT: The high-order polynomial runtime of L-WOT limits this implementation to smaller datasets. However, we see that even in experiments with 1000 samples, we are still able to run L-WOT in a reasonable time (for specifics, see experimental below).

Importantly, with a fixed small number of scales $S$, the runtime of E-WOT is the same as GW-OT as $n$ tends to infinity.

To demonstrate that E-WOT and GW-OT (Peyré et al., 2016) have comparable runtimes in practice, we added a new experiment that records the runtimes of our implementations (E-WOT and L-WOT) and GW-OT as we increase the number of samples in a dataset (we set the same following hyperparameters for all baselines in our implementations: $|S| = 20$, entropic regularization epsilon=1e-2, # sinkhorn iterations=100, distance matrix=euclidean) in seconds. In wall time in seconds/CPU time in seconds, we ran GW-OT and E-WOT on any given set of hyperparameters that run more than 20,000 seconds in wall time, we stop and replace its value with OOT (out of time). These times include the distance matrix calculation + spectral graph wavelet construction (if applicable) + optimizing the transport plan and were all run on the same NVIDIA RTX A6000 machine.

|       | n=100 | n=200 | n=500 | n=1,000 | n=5,000 | n=10,000 |
|-------|-------|-------|-------|---------|---------|----------|
| GW-OT | 46.468 / 0.933 | 86.899 / 3.476 | 277.421 / 4.401 | 425.596 / 6.737 | 3369.758 / 314.841 | 8928.406 / 917.152 |
| E-WOT | 150.134 / 3.031 | 189.133 / 4.622 | 267.743 / 5.397 | 462.553 / 10.340 | 5626.629 / 485.166 | 13510.435 / 1079.166 |
| L-WOT | 210.732 / 7.092 | 281.102 / 8.677 | 369.654 / 7.529 | 650.366 / 30.059 | 16220.132 / 1165.671 | OOT |

From these runtime benchmarks, we can see that GW-OT and E-WOT scale similarly in time with the number of samples in a dataset; the negligible time gap between GW-OT and E-WOT arises from the additional $|S|$ scales, but we expect that lowering the $|S|$ will likewise shrink the time gap. However, L-WOT explodes in runtime and may not be appropriate for aligning datasets with $n > 5000$.

We have shown that both E-WOT and L-WOT surpass the alignment quality of GW-OT in many experimental cases. Particularly since E-WOT and GW-OT have similar runtimes, we believe that it is compelling to use E-WOT over GW-OT in most cases. Even in the case of L-WOT, the increased quality of alignment may be worth the tradeoff in increased runtime. It is up to the user to select the most appropriate method for their alignment task.

Lastly, we would like to emphasize that the WOT framework itself does not inherit any runtime constraints, but rather it is the implementations and optimization methods like E-WOT and L-WOT that provide the explicit runtime complexity. Much like how GW-OT started with a naive implementation (Mémoli, 2011), but now has more efficient implementations based on entropic regularization (Peyré et al., 2016), WOT similarly aims to introduce a flexible framework for ML practitioners to align noisy and non-isometric datasets that are not explicitly tied to a specific implementation. We hope that our work opens up an exciting direction for future implementations and optimization techniques of WOT that are more efficient than E-WOT and L-WOT.

## F METRICS

We include brief descriptions of the metrics used in each experiment for completeness.

**FOSCTTM** (Experiment 1). Fraction of Samples Closer Than the True Match, introduced by Liu et al. (2019), is a measure of alignment error between two datasets. It quantifies the proportion of samples in one dataset that are closer to a given sample in the other dataset than its true match, averaged across all samples in both datasets.

To compute the FOSCTTM score for dataset A and B, we follow these steps:

1. For each sample point in dataset A, calculate the Euclidean distances between that point and all the data points in dataset B that have been projected into the dataspace of A (i.e. using barycentric projection).

2. Using these calculated distances, determine the fraction of projected samples in dataset B that are closer to the fixed sample point than its true match (i.e., the corresponding point in the second dataset that should be aligned with the fixed sample point).

3. Repeat steps 1 and 2 for all sample points in dataset A and take the average; the final value is the FOSCTTM score between A and B when B is projected into dataset A (note that this score is not equivalent to when A is projected into dataset B)

4. Perform steps 1-3 for each sample point in dataset B, calculating the distances to all points in dataset A projected into dataset B and determining the fraction of samples closer than the true match.

5. Finally, compute the average of the fractions obtained in steps 3 and 4 across all samples in both datasets to obtain the final FOSCTTM score.

The FOSCTTM score ranges from 0 to 1, with a perfect alignment resulting in a score of 0. In other words, when all samples are closest to their true matches, the average FOSCTTM will be zero. As the alignment quality decreases, the FOSCTTM score increases, indicating a higher fraction of samples that are closer to other points than their true matches.

**Relative Geodesic Error** (Experiment 2). This metric is calculated as

$$\epsilon(\mathbf{a}_i) = d_Y(\mathbf{a}_i, \mathbf{b}_i)/\text{area}(B)^{\frac{1}{2}}$$

where $\mathbf{a}_i$ is the projected sample from animal $A$ onto animal $B$. This projected sample is then compared with the ground truth sample $\mathbf{b}_i$ from animal $B$. $d_Y$ represents the geodesic distance on the animal which is calculated using a KNN approximation. The area over the animal is calculated using Delaunay triangulation as the surfaces.

**Label Transfer Accuracy** (Experiment 3). Introduced in Cao et al. (2020), Label Transfer Accuracy (LTA) is used when ground truth pairings are not available, and it evaluates how well cell types cluster together after alignment. It works by splitting the aligned data in half, training a kNN classifier on one half, and testing its accuracy on the other half. The classifier tries to predict cell types based on their proximity in the aligned space. Higher scores mean the alignment has grouped similar cell types closer together, allowing for more accurate predictions. This indicates a better quality alignment, where cells of the same type are consistently found near each other.

# G LIMITATIONS

While the Wavelet Optimal Transport (WOT) framework has demonstrated notable strengths in the domain of unpaired single-cell alignment and other noisy and non-isometric matching experiments, there are some limitations worth addressing:

**Scalability**. It is unclear whether WOT and its implementations can be applied directly to very large-scale datasets without some computation reduction strategies like mini-batch OT (Nguyen et al., 2022) as shown in Appendix E. Specifically, as the field of single-cell biology continues to expand and produce larger datasets, it will be crucial for future implementations of WOT to consider strategies for scalable alignment without compromising accuracy.

**Hyperparameters**. The wavelet kernel, scale aggregation operation, and entropic regularization are deeply coupled. Having some prior knowledge or validation set to select the optimal values for these hyperparameters would be ideal. However, in unpaired settings like ours, the key hyperparameter

that requires tuning is entropic regularization $\epsilon$ (this issue was similarly seen in Gromov-Wasserstein OT). Either by using a heuristic like the one proposed in Section 4 or another approach like Demetci et al. (2022a), readers must ensure that the selected $\epsilon$ does not result in a uniform transport plan (i.e. failed to converge to an informative plan).

Furthermore, we see a high variance in the experimental results between the two implementations, E-WOT and L-WOT. While these different instantiations of WOT offer unique filtering methods, the discrepancy in results suggests that there might be inherent complexities or nuances in the datasets that one method captures better than the other. This variability highlights the need for a deeper exploration into which filtering method (entropy-based or learned) and which types of wavelet kernels are more suited for specific types of datasets

**Performance in Low Noise Settings**. Our experiment results indicate that while WOT exhibits superior performance in scenarios with high noise, dropout, and non-isometry, it does not consistently outperform Gromov-Wasserstein OT in low-noise settings. This suggests that the benefits of using WOT will be more clear in situations with substantial technical variability, rather than in cleaner datasets. Readers should be aware of this trade-off when choosing the alignment technique best suited to their dataset's characteristics.

