# OpenReview forum: "Unpaired Single-Cell Dataset Alignment with Wavelet Optimal Transport"
_ICLR.cc/2025/Conference — Submitted to ICLR 2025_

### Official Review · Reviewer_Txuv · 2024-10-28

**Soundness:** 2
**Presentation:** 2
**Contribution:** 2
**Rating:** 5
**Confidence:** 3

**Summary:**

This paper introduces Wavelet Optimal Transport (WOT) for aligning unpaired single-cell datasets across different modalities. The core contribution lies in using spectral graph wavelets to decompose dataset signals into multiple scales and using filters during optimal transport. The authors propose two variants: E-WOT (entropy-based filtering) and L-WOT (learned filtering). This framework is claimed to generalize Gromov-Wasserstein and shows better performance on noisy data alignment tasks. Various experiments are performed on synthetic data, shape correspondence tasks, and real single-cell multi-omics datasets.

**Strengths:**

S1: The paper addresses a well-motivated problem (dataset alignment in single cell technologies). The proposed method outperforms many baselines in several setups.

S2: Multi-scale aggregation is a valid approach which has connections to Gromov-Wasserstein under specific conditions (Remark 1). The framework itself is general enough to allow a broad design space with many things to tune for improved performance (kernels, filter, scale aggregation, graph construction...)

**Weaknesses:**

W1: From a high-level perspective, the method combines established concepts (SGW and Gromov Wasserstein with filtering approaches). While the combination is novel, there appears to be limited theoretical development to help understand why and when the approach works well. Additionally, there are multiple moving parts that would benefit from a more detailed ablation study to shed light on different design choices. For instance,

a) Comparing at multiple scales is the primary motivation for this approach. I think it would be helpful to show (e.g., in experiments) which scales are most relevant and how much they contribute to performance. This is particularly important given the non-negligible performance variability between different WOT variants. In some experiments, E-WOT and L-WOT show varying performance (sometime underperforming some baselines). Without understanding this variability, it is challenging to predict when WOT will outperform existing methods or which variant to use.

b) This may be a minor point, but there are limited discussions of when to use which aggregation method (sum, max, potentially other weighting schemes) and how this choice affects results.

The overall impression is that there are many tunable components (scale selection, filtering approach, aggregation method, kernel choice) without sufficient guidance or understanding of their interactions.

W2: a) L-WOT may require some discussion on when or whether it converges at all. b) Is entropy always a good heuristic to emphasize informative scales? Using KDE with tunable bandwidth seems to introduce further complexity (in the Appendix, the authors report fixing Gaussian KDE bw at 0.4; have the authors considered adaptive bandwidth like Scott's or Silverman's?)

W3: It is not clear whether the experiments exclusively use the Chebyshev polynomial approximation. The paper would be strengthened if there is some analysis/comparison between the Chebyshev approximation vs the exact computation to provide some understanding of the trade-off between speed and performance.

W4: I think that runtime/memory benchmarking for all methods should be discussed more thoroughly (beside what is provided in Appendix E). What are the runtime for each method in each experiment? Moreover, constructing large, fully-connected graphs is expensive. It helps to understand the computational/memory cost when evaluating different methods for practical applications.

**Questions:**

Q1: Why is this work not positioned as a general framework for handling noisy, non-isometric spaces (many of which the authors leave as future directions)? Are there inherent limitations to it when applied to other domains? It is true that single-cell data provides an important application, but if the method is more general then I think it may be a good idea to broaden the experiment settings to truly evaluate its effectiveness against other baselines.

Q2: The authors state they cannot conduct hyperparameter tuning due to the unpaired setting (with some related details in Appendix C), is this a standard practice to pretend we don't have ground truth for tuning in these domains? Since there are many things to tune, how practical is the approach compared to the baselines? If we would like to avoid falling back to using heuristics, then how common is prior scale knowledge in practical settings? (In the paper, the authors cite Deutsch et al. (2016) but it seems generic and is not clear what scale is considered noise since there are no relevant experiments)

---

> ### Author Response · Authors · 2024-11-18
>
> We thank the reviewer for their time, effort, and constructive feedback. We address the reviewer’s concerns and questions below:
> ***
> **Comparing at multiple scales is the primary motivation for this approach. I think it would be helpful to show (e.g., in experiments) which scales are most relevant and how much they contribute to performance.**
>
> Please view our response to this question in the “Response to Common Questions & Concerns” comment.
> ***
> **There are limited discussions of when to use which aggregation method (sum, max, potentially other weighting schemes) and how this choice affects results. The overall impression is that there are many tunable components (scale selection, filtering approach, aggregation method, kernel choice) without sufficient guidance or understanding of their interactions.**
>
> Please view our response to this question in the “Response to Common Questions & Concerns” comment.
> ***
> **L-WOT may require some discussion on when or whether it converges at all.**
>
> Since L-WOT is a bilevel optimization problem where both the inner and outer loops are nonconvex, it is difficult to make any rigorous statements about the convergence of L-WOT. However, in practice, we find that limiting the number of outer loops to 100 is sufficient to obtain good performance in experiments.
> ***
> **Is entropy always a good heuristic to emphasize informative scales?**
>
> We **added a new section (Appendix 5.2.2)**, which explores the different wavelet scales and the informative ones. Specifically, in subsection “Filter Scales” of Appendix 5.2.2, we find that entropy closely matches the ideal filter for both single-cell datasets, demonstrating that is a good heuristic at least for scGEM and SNARE-seq.
>
> However, heuristics by definition can not always be good in all scenarios; the entropy heuristic is no exception. For instance, if one scale is uniformly distributed at random, the entropy heuristic would emphasize this scale even though it is complete noise. In practice, as shown in the experiments, we still obtain good results with this heuristic even if there are potential cases in which it may fail.
> ***
> **Using KDE with tunable bandwidth seems to introduce further complexity (in the Appendix, the authors report fixing Gaussian KDE bw at 0.4; have the authors considered adaptive bandwidth like Scott's or Silverman's?)**
>
> We did not try adaptive bandwidths for the Gaussian KDE since each dataset is unit normalized.
> ***
> **It is not clear whether the experiments exclusively use the Chebyshev polynomial approximation. The paper would be strengthened if there is some analysis/comparison between the Chebyshev approximation vs the exact computation to provide some understanding of the trade-off between speed and performance.**
>
> Our implementation exclusively uses the Chebyshev polynomial approximation, as stated in Section 3.1. This approximation is well-established in spectral graph wavelets literature [1] providing extensive theoretical guarantees and empirical validation. Since our work builds upon this foundation, we believe that re-validating the Chebyshev approximation would not substantially strengthen our contribution or provide new insights beyond what is already established in the literature. Our experimental results across multiple settings demonstrate that using the Chebyshev approximation effectively serves our method objective in handling noise and non-isometry in single-cell dataset alignment.
> ***
> **I think that runtime/memory benchmarking for all methods should be discussed more thoroughly (beside what is provided in Appendix E). What are the runtime for each method in each experiment? Moreover, constructing large, fully-connected graphs is expensive. It helps to understand the computational/memory cost when evaluating different methods for practical applications.**
>
> We believe the runtime/memory benchmarking is comprehensively covered in Appendix E where we provide
> 1. A detailed comparative table showing how each method scales with respect to feature dimensionality, number of samples, number of scales, and choice of wavelet kernels
> 2. Explicit timing benchmarks comparing our methods against GW-OT across different dataset sizes (from n=100 to n=10,000)
>
> Regarding fully connected graphs, this is a preprocessing step for computing intra-dataset distances and is not part of the core optimal transport algorithm - both our method and baselines require this distance computation.
>
> However, if there are specific aspects of the computational analysis you would like us to elaborate on, we would be happy to clarify those sections.
> ***
> **Why is this work not positioned as a general framework for handling noisy, non-isometric spaces (many of which the authors leave as future directions)? Are there inherent limitations to it when applied to other domains?**
>
> Please view our response to this question in the “Response to Common Questions & Concerns” comment.
> ***

---

> > ### Author Response · Authors · 2024-11-18
> >
> > (continued)
> >
> > **The authors state they cannot conduct hyperparameter tuning due to the unpaired setting (with some related details in Appendix C), is this a standard practice to pretend we don't have ground truth for tuning in these domains? Since there are many things to tune, how practical is the approach compared to the baselines? If we would like to avoid falling back to using heuristics, then how common is prior scale knowledge in practical settings?**
> >
> > For completely unpaired settings like single-cell dataset alignment, not having a validation set is indeed standard practice and reflects real-world scenarios where paired data is unavailable or impossible to obtain. This is not about "pretending" we do not have ground truth, but rather it reflects the genuine constraints of these biological applications.
> >
> > Regarding practicality, our method remains competitive with baselines even without extensive tuning. Most hyperparameters are fixed to default values (as detailed in Appendix C), and the few variable ones (like entropic $\epsilon$ regularization) are selected through an unsupervised procedure that we have now made explicit in Algorithm 2. In fact, our method achieves better results compared to baselines that face similar hyperparameter complexity.
> >
> > While prior scale knowledge is not commonly available in practice for single-cell data, our **newly added empirical analysis in Appendix D.2.2** shows that smaller-scale wavelets typically better reveal samples that should be aligned, while larger scales tend to muddle the data points. This pattern was consistent across both scGEM and SNARE-seq datasets, suggesting that even without prior knowledge, there are patterns in how different scales contribute to alignment quality, which guides interpreting informative scales even without prior knowledge.
> >
> > Lastly, we have improved the clarity of our unsupervised hyperparameter tuning by replacing the text description of our hyperparameter selection with Algorithm 2.
> > ***
> > [1] Wavelets on graphs via spectral graph theory. Applied and Computational Harmonic Analysis, 2011.

---

### Official Review · Reviewer_tWjN · 2024-11-02

**Soundness:** 3
**Presentation:** 2
**Contribution:** 2
**Rating:** 5
**Confidence:** 5

**Summary:**

This paper introduces the Wavelet Optimal Transport (WOT) method, a framework for aligning unpaired single-cell datasets from different modalities. WOT leverages spectral graph wavelet coefficients to capture multi-scale relationships within data and improve alignment robustness against noise, dropout, and non-isometric between data spaces. The authors present two implementations: E-WOT, using entropy heuristics to filter out irrelevant scales, and L-WOT, which dynamically learns the filters to improve alignment adaptivity. Experiments further demonstrate the WOT's superiority in both simulated and real scenarios.

**Strengths:**

1. Empirical Effectiveness. WOT consistently outperforms SOTA methods in high-noise scenarios, non-isometric conditions, and dropout cases across synthetic and real single-cell multi-omic datasets.
2. Theoretical Novelty. Using spectral graph wavelets for multi-resolution dataset alignment offers a novel view, and further theoretical rigor by generalizing GW-OT and ensuring robustness.
3. Flexibility. WOT allows different filters and optimization strategies to enhance performance based on dataset-specific characteristics.

**Weaknesses:**

1. Few misspellings in the main text (e.g., in line 012, 'throughput' should be 'throughout').
2. Figure 1 lacks clarity in illustrating both the task and framework.
3. Lack of comparisons with more recent single-cell paired alignment methods such as Harmony and scDML. They are the most common methods for single-cell alignment and deserve to be discussed. Additionally, the description of the weakness of related works in Section 2.1 remains for more detailed analysis about these methods.
4. The study lacks a broader range of experimental scenarios involving real single-cell multi-omic datasets. Results would be more convincing with diverse single-cell multi-omic datasets from different tissues and sequencing technologies, to demonstrate the method's effectiveness across various empirical conditions.
5. The proposed method is designed for two datasets' alignment, which restricts its applicability when aligning multiple datasets simultaneously. In fact, alignment of multiple samples may be a more common scenario.
6. The fixed hyperparameter setting is limited in applicability across real scenarios. Although the authors provide some advice in hyperparameter selection, utilizing an adaptive strategy is encouraged. Furthermore, even if it is difficult to give an adaptive strategy, sensitivity analysis of hyperparameters should be provided to show that this method can achieve good performance in most cases.

**Questions:**

1. In Sections 4.1 \& 4.2, task-specific baseline methods should be included.
2. How does your framework handle high-dimensional single-cell datasets? Common single-cell dataset dimensions are above 20K. Even if feature selection is performed, the most common input is 1-3K genes.
3. In Figure. 8, I can see that this method can keep the cell types separate. But I am more confused whether the SNARE-seq and ATAC-seq data have been mixed together successfully (need to color based on the dataset).
4. It seems to me that this method is generally applicable, and I wonder why it should be limited to the alignment of single cells? In fact, I think that if this method cannot effectively solve the problems unique to single-cell alignment, such as alignment of high dimensions and multiple datasets, it would be better to use public datasets for comparison and emphasize the general applicability of this method.

---

> ### Author Response · Authors · 2024-11-18
>
> We thank the reviewer for their time, effort, and constructive feedback. We address the reviewer’s concerns and questions below:
> ***
> **Few misspellings in the main text (e.g., in line 012, 'throughput' should be 'throughout').**
>
> This is not a misspelling. High **throughput** technologies are instruments that can analyze and create data for a large batch of samples.
> ***
> **Figure 1 lacks clarity in illustrating both the task and framework.**
>
> Can you be more specific about what part of the figure lacks clarity?
>
> The task is visualized through two graphs (X and Y) with nodes that need to be matched across datasets, specifically highlighting how nodes C and D in X correspond to nodes 3 and 7 in Y.
> The framework is explicitly shown through:
> - Multiple layers representing different scales (S0, S1, S2...)
> - The spectral graph wavelet coefficients (ψ) that define relationships between nodes
> - A visual representation of how these multi-scale views contribute to the total cost C
> - The filter F that removes uninformative scales and wavelets
> The figure directly communicates our core concept: WOT finds matches between points by considering their relationships at multiple scales, illustrated through the layered representation and the mathematical formulation above.
>
> If you have more specific points for improvement, we would be happy to include them.
> ***
> **Lack of comparisons with more recent single-cell paired alignment methods such as Harmony and scDML.**
>
> In unpaired dataset alignment, the methods referenced in Table 2 are current state-of-the-art methods.
>
> Note that while many newer works may fall into the category of data alignment, they often have very strong assumptions between the data spaces and incorporate those assumptions into their method. For example, Harmony [1] assumes knowledge of cell type labeling or batch information. Other examples include assuming a prior knowledge graph between features [3], utilizing weakly linked features [2], and more [4,5]. In short, these methods are not truly "unpaired alignment" methods and thus would not be a fair and meaningful baseline. Additionally, these methods cannot operate on modalities such as brightfield imaging/scRNA-seq, etc., or other modalities outside of single-cell biology where these assumptions may not hold.
>
> Furthermore, scDML is a data clustering technique, not a cross dataset alignment technique. Specifically, it corrects for batch effect for datasets in the same space (i.e. given two RNA-seq datasets, cluster the two). This is a different task altogether than ours because we’re trying to align datasets **not** in the same space (i.e. given an RNA-seq dataset and an ATAC-seq dataset, align the two).
> ***
> **The description of the weakness of related works in Section 2.1 remains for more detailed analysis about these methods.**
>
> Our paper provides an analysis of related works' limitations through both general discussion and experiments:
> - Section 2.1 introduces existing methods and their general limitations
> - Section 4.1 explicitly demonstrates how baseline methods like GW fail in high-noise scenarios
> - Section 4.2 shows baseline limitations in handling non-isometric relationships
> - Real single-cell experiments in Section 4.3 validate these limitations in practice
>
> Rather than just stating weaknesses, we systematically demonstrate them through carefully designed experiments that isolate failure modes. The progression from controlled experiments to real data provides clear, empirical evidence of where and why existing methods fall short (noise and non-isometry). If the reviewer has specific aspects of the baseline methods they feel warrant deeper analysis, we welcome more detailed feedback.
> ***
> **The study lacks a broader range of experimental scenarios involving real single-cell multi-omic datasets. Results would be more convincing with diverse single-cell multi-omic datasets from different tissues and sequencing technologies, to demonstrate the method's effectiveness across various empirical conditions.**
>
> We agree that aligning samples across different tissues would be useful. However, our current choice of the scGEM and SNARE-seq datasets was deliberate and comprehensive for a couple of reasons:
> 1. These datasets represent distinct biological scenarios and technical challenges:
>     * scGEM captures a dynamic process (cell reprogramming) with gradual state transitions
>     * SNARE-seq represents discrete cell types with clear cluster boundaries
>     * They use different measurement technologies (gene expression/DNA methylation vs RNA/chromatin accessibility)
> 2. These datasets are well-established benchmarks in the field, enabling direct comparison with multiple baseline methods using standardized evaluation metrics.
>
> While additional datasets could provide further validation, our current experiments already demonstrate WOT's performance across significantly different biological contexts and technical conditions.
> ***

---

> > ### Author Response · Authors · 2024-11-18
> >
> > (Continued)
> >
> > **The proposed method is designed for two datasets' alignment, which restricts its applicability when aligning multiple datasets simultaneously. In fact, alignment of multiple samples may be a more common scenario.**
> >
> > We acknowledge this limitation, but it's important to note that this is not unique to WOT - it is an inherent challenge for all optimal transport-based alignment methods. The pairwise nature of optimal transport formulations makes multi-sample alignment fundamentally challenging.
> >
> > While extending WOT to handle multiple datasets simultaneously would be valuable future work, our current focus was on improving the quality of pairwise alignment, particularly in handling the noise and non-isometry challenges specific to single-cell data.
> > ***
> > **The fixed hyperparameter setting is limited in applicability across real scenarios. Although the authors provide some advice in hyperparameter selection, utilizing an adaptive strategy is encouraged. Furthermore, even if it is difficult to give an adaptive strategy, sensitivity analysis of hyperparameters should be provided to show that this method can achieve good performance in most cases.**
> >
> > Please view our response to this question in the “Response to Common Questions & Concerns” comment.
> > ***
> > **In Sections 4.1 & 4.2, task-specific baseline methods should be included.**
> >
> > Section 4.1 is a toy dataset specifically designed to isolate and evaluate noise handling capabilities. There are no task-specific baselines because this is a controlled experiment to demonstrate that WOT can maintain accurate alignment even in high noise regimes.
> >
> > Regarding Section 4.2 (shape correspondence), yes, there have been shape-matching specific methods that benchmark on the SHREC20 dataset. However, these methods can only operate on 3D shapes and cannot be scaled to higher dimension datasets like single-cell datasets - directly comparing WOT with these method would not be a fair and meaningful baseline. As such, we have chosen only to compare with other methods (OT-based like GW and non-OT based like UnionCom) that can scale to higher dimensions.
> > ***
> > **How does your framework handle high-dimensional single-cell datasets? Common single-cell dataset dimensions are above 20K. Even if feature selection is performed, the most common input is 1-3K genes.**
> >
> > WOT operates on the spectral graph wavelets (SGWs) derived from the pairwise distance matrices of the datasets, rather than directly on the high-dimensional data points themselves (Section 3.1). Hence, the dimensionality is primarily related to the computation of pairwise distances rather than the SGWs or the WOT algorithm itself.
> >
> > However, to directly address your concern, pairwise distance subroutines have been highly optimized in popular libraries like SciPy and PyTorch and have not been an issue in our experiments (in Section 4.1, each sample has 2000 dimensions.)
> > ***
> > **In Figure. 8, I can see that this method can keep the cell types separate. But I am more confused whether the SNARE-seq and ATAC-seq data have been mixed together successfully (need to color based on the dataset).**
> >
> > We **updated Figure 8 with a new plot** displaying samples colored by their respective dataset rather than cell type.
> > ***
> > **It seems to me that this method is generally applicable, and I wonder why it should be limited to the alignment of single cells? In fact, I think that if this method cannot effectively solve the problems unique to single-cell alignment, such as alignment of high dimensions and multiple datasets, it would be better to use public datasets for comparison and emphasize the general applicability of this method.**
> >
> > Please view our response to this question in the “Response to Common Questions & Concerns” comment.
> > ***
> > [1] Fast, sensitive and accurate integration of single-cell data with Harmony. Nature methods, 2019.
> >
> > [2] Integration of spatial and single-cell data across modalities with weakly linked features. Nature Biotechnology, 2023.
> >
> > [3] GLUER: integrative analysis of single-cell omics and imaging data by deep neural network. BioRxiv, 2021.
> >
> > [4] Single-cell multi-omic integration compares and contrasts features of brain cell identity. Cell, 2019.
> >
> > [5] Deep generative modeling for single-cell transcriptomics. Nature methods, 2018.

---

> > > ### Comment · Reviewer_tWjN · 2024-11-29
> > >
> > > Thank you for the author's detailed response and careful revision of the manuscript. After carefully reading the revised manuscript and other reviewers' opinions, some of my confusions were answered, but there are still some concerns that I think the author needs to further clarify. The following are the most important ones (note, not all questions, just those that I think need more response):
> > >
> > > 1. Regarding the author's view on Harmony, I am skeptical. First, although they are paired methods, they are the most commonly used alignment method in practice. Indeed, Harmony was proposed for alignment between scRNA-seq datasets, but this does not affect its use for alignment between scRNA-seq and scATAC-seq datasets. In addition, I listed these two methods mainly as examples. Even the methods suitable for alignment between scRNA-seq and scATAC-seq are still too numerous to mention. For example, scJoint (https://www.nature.com/articles/s41587-021-01161-6) can do this.
> > >
> > > 2. Experimental Fairness. It does seem that the method proposed by the author has a performance advantage over the unpaired method. The authors believe that it is unfair to compare paired and unpaired methods. In terms of methodology, I agree with this view. But in practice, I completely disagree. Because if our goal is to align scRNA-seq and scATAC-seq to the same space for downstream bioinformatics analysis, we don’t care whether to use the unpaired method or not. We only care about the quality of the alignment results. In particular, if unpaired methods perform much worse than paired methods, then we will have less motivation to use these methods, even if they have technical innovations. What I mean is that I don’t understand how this technical innovation has any performance advantage over the existing paired methods. And the existing methods may be able to be extended to align three or four datasets at the same time, while this method is limited.
> > >
> > > Overall, this paper is technically novel and interesting. But I still wonder what advantages it has in practice. And the methods for aligning across sequencing data seem to be very "crowded", which requires the authors to emphasize the differences between their proposed method and other methods, and not just limited to technical differences.
> > >
> > > In addition, I strongly agree with reviewer Txuv's opinion. While the paper appears to be technically very novel, it appears to lack breadth and depth. In terms of content, I think the author needs to continue to work hard to improve these shortcomings or answer my concerns. Therefore, I will maintain my original score for now. Looking forward to further response from the author.

---

> > > > ### Author Response · Authors · 2024-11-29
> > > >
> > > > Thank you for the reviewer's continued discussion of our work. We respectfully disagree with the assessment of practical applicability and would like to clarify some important points:
> > > >
> > > > (1) The fundamental issue is that paired samples or prior labeling information (like cell-type labeling) are challenging or even impossible to obtain in many real-world scenarios. In limited cases where this information is available, it is obvious that one should leverage this prior information for better alignment. However, _in most practical cases_, it is not a matter of deciding between paired or unpaired methods, but rather, one can **only** use unpaired methods.
> > > >
> > > > For example, in modalities like spatial transcriptomics, single-cell metabolomics, and single-cell glycomics, obtaining definitive cell type labels is nearly impossible due to technical limitations and the complex nature of these measurements. Even in more established modalities like scRNA-seq and scATAC-seq, obtaining cell-type labeling (used as input for scJoint) requires additional complex computational or experimental procedures such as manual expert annotation, automated tools like SingleR, or extensive marker gene analysis. As a result, not all scRNA-seq or scATAC-seq datasets have cell type labeling information. In these common cases where prior information is not provided, one can still use WOT but cannot use works like scJoint. Therefore, in practice, our method is much more applicable to real problem settings.
> > > >
> > > > (2) Beyond the issue of requiring often unattainable prior information, existing methods are typically restricted to specific modality pairs. For instance, scJoint can only align between scRNA-seq and scATAC-seq, making it unsuitable for other experimental datasets like scGEM (scRNA-seq and DNA methylation). As demonstrated in our single-cell experiments, WOT can align multiple modalities and is theoretically designed to align any modality combination. While we acknowledge that there are inherent tradeoffs between generality and specificity in any method, we have intentionally developed WOT as a general-purpose solution that can serve as a "swiss-army knife" for any type of modality alignment task (as described in our introduction).
> > > >
> > > > Furthermore, we must correct a misunderstanding about Harmony's capabilities. Harmony cannot align between different modalities, including scRNA-seq and scATAC-seq. As explicitly stated in its methods section and "Assumptions about Input Data," Harmony was designed specifically for aligning scRNA-seq data. Consequently, it cannot be applied to our single-cell datasets, which require alignment between different modalities.
> > > >
> > > > In summary, we want to emphasize that the framing of choosing between paired and unpaired methods is misleading. While paired information should certainly be leveraged when available, the reality is that most real-world situations are inherently _unpaired_. In these cases, there is no decision to make between paired and unpaired methods - unpaired methods are the only viable option. As such, we believe that WOT is a practically useful method that can align between any pair of single-cell modalities in a completely unpaired manner.

---

### Official Review · Reviewer_oyEN · 2024-11-03

**Soundness:** 2
**Presentation:** 1
**Contribution:** 2
**Rating:** 5
**Confidence:** 2

**Summary:**

The authors present Wavelet Optimal Transport (WOT) based on spectral graph wavelets to align different single-cell datasets, which can be challenging due to noise, dropout, and batch effects. By introducing two versions of WOT, namely L-WOT and E-WOT, the authors demonstrate that these methods are collectively better the the state-of-the-art methods on two simulation and two real single-cell datasets.

**Strengths:**

The authors address important problem of aligning unpaired single-cell datasets, which seems to not have been addressed in the literature previously. While many unpaired methods exist in different applications, such as image-to-image translation as the authors rightly point out, single-cell context requires additional careful considerations. The mathematical details are thorough and the algorithm itself is familiar to the literature, as it relies on projected gradient descent and/or alternating minimization schemes. I think when the paper is put in right format and much more user-friendly writing style, it will be a great contribution to the community

**Weaknesses:**

The biggest weakness of this paper is the **presentation**. I read it through multiple times and it was still quite confusing and not sure who the audience is. This is clearly not going to be the computational biology practitioners as the paper is too technical without providing connections and insights sufficiently to the single-cell applications. It felt as though the authors had the idea of WOT on spectral graph wavelets first and then tried to find for some relevant applications afterwards, which led to awkward connections. The experiments felt a bit rushed in that the main highlights (Section 4.3) are only dedicated half a page, without clear and thorough investigations on why the WOT algorithms might be performing better (I also didn't like the practice of "taking" the numbers from other papers).

- It was not clear what alignment of "unpaired" single-cell dataset would actually mean. What would be the use case for this, how would biologists benefit from such alignment for scientific research? For image-to-image translation, yes there are clear reasons, but for alignment of single-cell dataset, it was not so clear from the paper
- The authors need to keep referring back & connecting to the single-cell example/application throughout the presentation of the algorithm. Section 3 is almost entirely separated from the single-cell application, and it is unclear how these two are related. For instance, the "scale" is discussed a lot in the algorithm, but what does scale actually mean in single-cells? The wavelets can provide a basis for decomposition in spectral & temporal domain, but what does it actually mean for single-cell images?
- I think the experimental results are not presented in reader-friendly manner. The evaluation metrics keep changing (geodesic, label transfer accuracy), which the practitioner not familiar with them would have hard time understanding - What do each of these metrics really tell you? Wouldn't it be also important to discuss what filters are used and learned for L- and E-WOT in these real-world examples? What do each of baseline in real-world example do (SCOT, UNIONCOM, etc..) and how/why WOT methods are better as we see in the table?
- "L-WOT performs much better than E-WOT and existing methods on the scGEM while the inverse is seen in SNARE-seq. A potential reason for this difference is that scGEM profiles cells in dedifferentiation, so the boundaries of cell types are not as clear as those of SNARE-seq ~" => This feels like scratching the surface and not really getting at the reason why one might perform better than the other
- Some illustrative (quantiative and qualitative) figure on the experiment results for 4.3 would be informative.

All in all, the authors should focus much less on the mathematical underpinnings but focus more on biological aspect for this to be a valid contribution to the community.

**Questions:**

- The simulation dataset seems to have dimension of from 1,000 to 2,000, but the real-world single-cell datasets remain at 10~30. Could you explain if this is common combination and how this might affect the algorithm?

---

> ### Author Response · Authors · 2024-11-18
>
> We thank the reviewer for their time, effort, and constructive feedback. We address the reviewer’s concerns and questions below:
> ***
> **I read it through multiple times and it was still quite confusing and not sure who the audience is. This is clearly not going to be the computational biology practitioners as the paper is too technical without providing connections and insights sufficiently to the single-cell applications.**
>
> We respectfully disagree that the paper lacks clear motivation and connection to single-cell biology. Our method was specifically designed to address two critical challenges unique to single-cell data alignment: high technical noise and non-isometric relationships between modalities. These are not arbitrary technical innovations looking for applications - they directly respond to fundamental challenges in the single-cell field as detailed in our introduction.
>
> Unlike traditional GW methods that were made for isometric matching, WOT was intentionally made to handle the complex realities of single-cell data. The progression of our experiments deliberately and consistently demonstrates this connection back to single-cell: we first (1) isolate and validate WOT's ability to handle noise (bifurcation experiment) and non-isometry (shape correspondence experiment) before (2) showing its effectiveness in real single-cell datasets.
>
> We also respectfully disagree that the paper's technical depth makes it inaccessible to computational biology practitioners. Computational biology, particularly single-cell analysis, regularly uses technically challenging mathematical methods from other fields - from manifold learning in trajectory inference to probabilistic modeling. Our target audience is precisely these practitioners.
>
> Additionally, much like how GW was initially developed for object matching but has now been applied to even the field of single-cell [1], our method is initially developed for single-cell but can also be applied in future works to other domains; this general applicability is a benefit rather than a drawback.
> ***
> **It was not clear what alignment of "unpaired" single-cell dataset would actually mean. What would be the use case for this, how would biologists benefit from such alignment for scientific research?**
>
> As stated throughout the introduction and mathematically explicit in lines 158-161, unpaired alignment between single cell datasets means that we want to define the mappings between cells of one modality (like scRNA-seq) to cells of another modality (like ATAC-seq) without any apriori knowledge on the joint distribution of these modalities.
>
> Since each modality provides a different measurement (i.e. view) of the cell state, biologists often want to have data from multiple modalities from the same cell. However, this case is often impossible since many measurements are destructive, meaning you cannot conduct multiple measurements on the same cell. Hence, having unpaired single-cell alignment methods like WOT is important because it effectively provides data from multiple modalities on the “same” cell even when it is biologically impossible to measure these modalities on the same cell.
>
> ***
> **The authors need to keep referring back & connecting to the single-cell example/application throughout the presentation of the algorithm. Section 3 is almost entirely separated from the single-cell application, and it is unclear how these two are related. For instance, the "scale" is discussed a lot in the algorithm, but what does scale actually mean in single-cells?**
>
> We agree that some terms like “scale” in the methods section may be ambiguous in how they relate to single-cell. However, since Section 3 is our Methods section, we deliberately structured the paper to first establish the mathematical foundation of WOT before demonstrating its practical relevance to single-cell biology in the Experiment section. This is standard practice for methods papers, where a clean technical presentation enables readers to fully understand the approach before seeing its application.
>
> To clarify the meaning of scale, it intuitively represents the high or low frequency patterns of the single cells dataset. While there is not a direct a one-to-one correspondence with a specific biological variation, one could imagine the low-frequency scales (higher valued scales) corresponding to global, slow-moving patterns like cell type while high-frequency scales (lower valued scales) represent local, fast-moving patterns like noise. Each scale represents a different level of pattern resolution from the more global scale information to local scale information.  The goal of WOT is to separate the various scales of each single-cell dataset and align the most common scales, which directly reduces noise and non-isometry between different datasets.
>
> To further address these concerns, we **added a new section (Appendix D.2.2)** to visualize and attempt to explain the scales of wavelets for single-cell datasets.
> ***

---

> > ### Author Response · Authors · 2024-11-18
> >
> > (continued)
> >
> > **I think the experimental results are not presented in reader-friendly manner. The evaluation metrics keep changing (geodesic, label transfer accuracy), which the practitioner not familiar with them would have hard time understanding - What do each of these metrics really tell you?**
> >
> > We agree and have **added Appendix E** which summarizes each metric. However, please note that different evaluation metrics are necessary since each experiment is different and necessitates a different evaluation for meaningful insights.
> > ***
> > **Wouldn't it be also important to discuss what filters are used and learned for L- and E-WOT in these real-world examples?**
> >
> > Section 3.3 and 3.4 outline how WOT derives the filters in E-WOT and L-WOT. Additionally, we have **added a new section (Appendix D.2.2)** that visualizes and analyzes the filters from L-WOT and E-WOT to address this concern.
> > ***
> > **What do each of baseline in real-world example do (SCOT, UNIONCOM, etc..) and how/why WOT methods are better as we see in the table?**
> >
> > Each baseline method is a dataset alignment method: they find a mapping from samples in one dataset to samples in another dataset. As discussed in our method and experiment section, WOT performs better because we explicitly model the various scales of each dataset and filter out the noisy and non-isometry components. This robustness in the presence of noise and non-isometry (as demonstrated in our experiments) is likely why WOT performs better in various experiments.
> > ***
> > **"L-WOT performs much better than E-WOT and existing methods on the scGEM while the inverse is seen in SNARE-seq. A potential reason for this difference is that scGEM profiles cells in dedifferentiation, so the boundaries of cell types are not as clear as those of SNARE-seq ~" => This feels like scratching the surface and not really getting at the reason why one might perform better than the other**
> >
> > We conducted an **additional experiment** to visualize the filter weights and wavelet scales for the single-cell experiments in **Appendix D.2.2** which may provide some intuition on why we see this differing performance. However, please understand that a rigorous conclusion to answer why one heuristic performs better than another on a specific dataset is incredibly non-trivial in machine learning. A similar analogy is trying to understand why a learning rate of 1e-2 is better than 1e-3 for one experiment but worse for another experiment.
> > ***
> > **Some illustrative (quantiative and qualitative) figure on the experiment results for 4.3 would be informative.**
> >
> > We added a **new section (Appendix D.2.2)** which provides more illustrations for Experiment 4.3. Please also view the beginning of Appendix D.2 for visualizations of single-cell dataset alignments. If you have any specific suggestions, we would be happy to include them.
> > ***
> > **The authors should focus much less on the mathematical underpinnings but focus more on biological aspect for this to be a valid contribution to the community**
> >
> > We respectfully disagree. As a submission to ICLR, the paper needs to balance mathematical novelty with biological applications. The technical depth is not excessive - it is essential for demonstrating and explaining the method's contributions to the machine-learning community. Through experiments and direct motivation by problems in single-cell, we also balance this technical depth with substantive relevance and contribution to the field of single-cell.
> > ***
> > **The simulation dataset seems to have dimension of from 1,000 to 2,000, but the real-world single-cell datasets remain at 10~30. Could you explain if this is common combination and how this might affect the algorithm?**
> >
> > Real single-cell datasets like scRNA-seq are commonly in the thousands to tens of thousands of dimensions. Datasets of 10-30 dimensions are only after dimensionality reduction. WOT operates on the spectral graph wavelets (SGWs) derived from the pairwise distance matrices of the datasets, rather than directly on the high-dimensional data points themselves. The construction of the pairwise distance matrices, which serve as the input to the SGW transform, is indeed affected by the dimensionality of the data. As the number of dimensions increases, the computation of pairwise distances becomes more challenging due to the curse of dimensionality. High-dimensional spaces tend to be sparse, and the notion of distance becomes less informative as the dimensionality grows.
> >
> > However, once the pairwise distance matrices are computed as a preprocessing step, the subsequent steps of constructing the SGWs and applying the WOT algorithm are not explicitly affected by the dimensionality of the original data points. The SGWs are derived from the eigendecomposition of the graph Laplacian, which is constructed purely based on the pairwise distance matrices.
> > ***
> > [1] SCOT: single-cell multi-omics alignment with optimal transport. Journal of computational biology, 2022.

---

### Official Review · Reviewer_1Hb6 · 2024-11-09

**Soundness:** 2
**Presentation:** 3
**Contribution:** 3
**Rating:** 5
**Confidence:** 4

**Summary:**

The authors propose a new method, "Spectral Graph Wavelet Optimal Transport" (WOT), to align unpaired single-cell multi-omic datasets. To do so, they perform Gromov-Wasserstein OT alignment between two domains (-omic measurement types) but unlike existing methods, intra-domain distances are computed based on spectral graph wavelets in order to capture multi-scale structural information on the graphs constructed on each modality. They include scale-varying filters in their framework in order to filter out uninformative signals (or noise). There are two proposed strategies for choosing filters: (1) heuristically choosing them based on the entropy of the wavelets in each scale, as estimated based on kernel density estimation and (2) learned via an alternating optimization procedure, where the transport plan and the filters are alternatingly optimized via Sinkhorn iterations and stochastic gradient ascent, respectively.

Overall, the approach is well-motivated and the proposed method is novel. While computational complexity is a challenge, experiments demonstrate the benefits in non-isometric and noisy settings. However, I have several remaining questions after reading the paper, especially around how these experiments and baselines are set up. Most importantly, the lack of sufficient information around the hyperparameter selection worries me about the fairness and quality of benchmarking experiments. My current score of 3 is mostly based on this. I am happy to update my score and recommend acceptance if the authors satisfactorily address this concern in the discussion period.

**Strengths:**

The paper is well-written (other than missing details) and easy to follow and the problem is well-motivated. Experimental section includes a number of simulated and real-world datasets that cover a mix of scenarios and the results are compared to relevant baselines.

**Weaknesses:**

**Updated Review** After the rebuttal process, seeing the description of the unsupervised hyperparameter tuning procedure, I am updating my score from 3 to 5, as it partially addresses my concern from #1 below. Having said so, it appears that the hyperparameter tuning procedure does not account of hyperparameters like \rho, which will matter in real-world unpaired dataset integration, as these datasets tend to contain disproportionate cell type representation. Overall, while I think the authors make a contribution towards improving OT-based single-cell multi-modal data alignment (a challenge with scientific impact), I believe additional work needs to be done on **real-world unpaired datasets** before wrapping up this work.
I keep my original review below for reference.

**1.** The largest concern I have with this submission is the lack of information around hyperparameter selection in benchmarking experiments. The proposed method has a high number of hyperparameters that require tuning:
- (1) choice of k for the initial kNN graph,
- (2) the bandwith \sigma for the RBF kernel that forms the weighted adjacency matrices,
- (3) scale parameters [0...S],
- (4) \epsilon for the entropic regularization of OT,
- (5) choice of aggregation function {min, max, sum},
- (6) choice of wavelet generating function g {low-pass heat kernel, tight-frame Meyer kernel, ...},
- (7) the hyperparameters associated with the chosen g function,
- (8) threshold variable \delta for L-WOT,
- (9) hyperparameters associated with KDE for E-WOT (bandwidth).

The authors emphasize that the experiments are conducted in "fully unpaired scenario, where **we don't have validation data to conduct hyperparameter tuning**". Then, in Appendix C, they report different sets of hyperparameters for each dataset. If the experiments weren't conducted with default parameters (i.e. default changes based on dataset) but neither was any validation data was used, then how were different sets of hyperparameters chosen for each dataset? Is there a heuristic / self-tuning procedure to adopt hyperparameters to each dataset without tuning with validation data? This information is lacking. I looked at Appendix C1 on "Guidance on Choosing Hyperparameters" but it does not explain the experiments: it describes keeping aggregation function fixed as summation while reported hyperparameters use a range of aggregation function and secondly, the only heuristic described for self-tuning is from Demetci (2022b), which would only account for k and \epsilon, leaving many other hyperparameter to tune. If there was a self-tuning procedure used, this would be crucial to describe in the paper. The results they compare against uses default parameters for UnionCom, Pamona, MMD-MA, Seurat, bindSC and a self-tuning heuristic for SCOT, SCOTv2 and cross-modal AE. If hyperparameters of E-WOT and L-WOT were, in fact, tuned using data from the experiments, then the correct numbers to compare against would be Fig 2 results from Demetci (2022a):

|   |SCOTv2 | SCOT | UnionCom | Pamona | MMD-MA | Cross AE | BindSC | Seurat |
| ---- |  ----  |  ----  |  ----  |  ----  |  ----  |  ----  |  ----  |   ----  |
**SNARE-seq** | 0.927 | 0.982 | 0.423 | 0.686 | 0.942 | 0.689 | 0.734 | 0.684 |
**scGEM**      | 0.643 | 0.576 | 0.582 | 0.651 | 0.588 | 0.523 | 0.449 | 0.423 |

**2.** (Minor point) I think the discussion of previous methods not explicitly reducing noise is not entirely accurate: "[previous methods] do not explicitly reduce dataset-intrinsic noise or signal". E.g. UnionCom, Pamona, SCOT, SCOTv2 all use kNN graphs (using only connectivity or Euclidean distance between PCs) on dimensionality-reduced data, which would reduce noise, albeit in a much more naive way than the proposed method here. Cross-modal autoencoders align datasets in a smaller latent space learned by autoencoders, which would be expected to have lower noise level, as well.

I have other questions and comments below, but the major questions I have are in point #1 above.

**Questions:**

**1.** How were the hyperparameters of SCOT, SCOTv2, UnionCom and Pamona chosen for the SHREC20 experiments? What were these hyperparameters?
**2.** What does "Multiple" mean for "Wavelet" parameter in Appendix C? How do you choose the set of scales?
**3.** When computing the kNN graph, what representation of data is used for single-cell datasets? Is it the raw gene expression data, normalized data, or some dimensionality-reduced version of the data? If it's dimensionality reduced, is it based on PCA, LDA, tSNE etc? Do you consider Euclidean distances or correlation or a different
**4.** What discrepancy measure do you use for L in Equation 4?
**5.** For the GW baseline used in bifurcation matching experiments, how are the intra-domain distances computed (i.e. what is the choice of d_A and d_B used from Eq 1 for this experiment)? Why are single-cell alignment baselines excluded from this experiment?
**6.** Have you attempted to compare methods in settings with validation data? Is the advantage from having a better self-tuning procedure or do you also outperform existing methods when validation data is available?

These are less major questions than the ones in weaknesses section but answers to 1-5 would be good to include in the paper for completeness and replicability.

---

> ### Author Response · Authors · 2024-11-18
>
> We thank the reviewer for their time, effort, and constructive feedback. We address the reviewer’s concerns and questions below:
> ***
> **Weakness 1**
>
> While we acknowledge that the range of flexibility with our framework can be initially daunting, we want to make clear that most hyperparameters can be fixed to default values as specified in Table 3 of Appendix C. There are only three hyperparameters that are variable between each experiment: (1) the entropic $\epsilon$ regularization, which is a hyperparameter common to any entropic OT method, (2) the aggregation operation, and (3) the weight normalization of the graph edges.
>
> We **added Algorithm 2 to Appendix C**, which summarizes the unsupervised hyperparameter procedure originally described at the beginning of Appendix C in our initial submission. We do not use result metrics from experiments to guide hyperparameter selection.
>
> This unsupervised procedure explains the different parameter values observed across datasets. Importantly, we default to sum aggregation and RBF normalization in most cases, only deviating when necessary according to Algorithm 2. The $\epsilon$ parameter shows the most variation, which aligns with previous work showing that fixed defaults often yield uninformative transport plans [1]
>
> Further, to address your concerns about how we initially selected the defaults for the hyperparameters, we **added Table 4**, describing where each value came from.
> ***
> **Question 1**
>
> We have written a **new section (Appendix C.2)** with details on hyperparameter selection for baselines.
> ***
> ​​**Question 2**
>
> Multiple refers to evaluating multiple kernels in the experiment. For instance, in Experiment 3, we evaluated both a simple tight kernel and the heat kernel. We have added this clarification in Table 5. The specific scales are determined by the wavelet kernel, which we default to the values in [3]
> ***
> **Question 3**
>
> We directly use the reduced-dimension single-cell datasets provided by [1] and [5]. For completeness, we summarize their procedure:
>
> - scGEM: both datasets are reduced to their corresponding dimensions through normalized PCA
> - SNARE-seq: RNA-seq was reduced through PCA and ATAC-seq was reduced through the topic modeling framework cisTopic [6]
> ***
> **​​Question 4**
>
> Thank you for pointing this out. We use the quadratic loss $L(a,b) := \frac{1}{2}(a - b)^2$. We have revised the beginning of the experiment section to include this important detail.
> ***
> **​​Question 5**
>
> For the bifurcation experiment, we use the Euclidean distance as the intra-domain distances for both GW and our method. We focused on comparing WOT with only GW-OT in the bifurcation experiment since GW-OT forms the theoretical foundation of WOT (as shown in Remark 1). This controlled comparison allowed us to specifically demonstrate the benefits of incorporating wavelets into the optimal transport framework, particularly in high-noise scenarios.
> ***
> **Question 6**
>
> Our work specifically focuses on the unpaired setting since it reflects the reality of single-cell data collection - paired samples are often impossible or prohibitively expensive to obtain. While we haven't evaluated performance with validation data, this was a deliberate choice aligned with our goal of developing methods that work effectively "out of the box" in real-world scenarios where paired samples are unavailable. Evaluating performance with validation data would be an interesting direction for future work, but it would not address the core unpaired challenge.
> ***
>
> [1] SCOT: single-cell multi-omics alignment with optimal transport. Journal of computational biology, 2022.
>
> [2] Large sample analysis of the median heuristic. 2017.
>
> [3] PyGSP: Graph Signal Processing in Python. 2014.
>
> [4] Unsupervised topological alignment for single-cell multi-omics integration. Bioinformatics, 2020.
>
> [5] MATCHER: manifold alignment reveals correspondence between single cell transcriptome and epigenome dynamics. Genome Biology, 2017.
>
> [6] cisTopic: cis-regulatory topic modeling on single-cell ATAC-seq data. Nature Methods, 2019.

---

> ### Comment · Reviewer_1Hb6 · 2024-11-21
> **Few questions on the self-tuning procedure**
>
> **Edit:** I previously posted this under the wrong comment, deleted it and moved it here.
>
> Dear authors,
>
> Thank you very much for responding to my questions and updating the document. I do believe single-cell multi-omic integration in a fully unpaired setting, as described in the paper, is an important task for scientific applications, so I think the details of self-tuning procedure are important. My initial worry was that the hyperparameters may have been chosen with some validation data (since the descriptions on this were missing), leading to unfair benchmarking, but it appears this is not the case, so I will incline towards raising my score. However, I have some additional questions and concerns regarding the self-tuning procedure in Algorithm 2 in Supplementary Section C. My understanding is that you are doing the following:
>
> 1. Start out by a very small $\epsilon$ value, $10^{-4}$, and use sum for aggregation function and RBF for norm. If this already gives a valid coupling, just use these hyperparameters.
> 2. But small $\epsilon$ values could give degenerate couplings, e.g. due to NaNs. In that case you gradually increase $\epsilon$ until you get a valid coupling (no NaNs) thats sufficiently different than a uniform coupling, "sufficiently different" described by $\eta$.
> 3. If you've raised $\epsilon$ up to $\epsilon$ $\geq 1$ and still don't have a valid coupling, you cycle through this procedure trying out different aggregation functions first, and then different norm values.
>
> **Based on this here are my questions:**
> **1.** Tending to small $\epsilon$s (i.e. stopping hyperparameter tuning at the first point of sufficiently high enough $\epsilon$ to get a valid coupling) makes sense for aligning datasets that are already paired (e.g. SNARE-seq, scGEM experiments from the paper). These datasets have underlying 1-to-1 mapping of cells since they were jointly measured so the ground-truth coupling is super sparse (I am aware you aren't using this info when solving the coupling, just using it for benchmarking). However, this likely won't be the case for real-world separately sequenced datasets since the underlying biological manifold won't have 1-to-1 matches between cells. In 1-to-many settings, higher $\epsilon$s would likely be more favorable, so this self-tuning procedure may not be as effective. I don't know if this is practically possible in the next 5 days of the rebuttal phase but: Could you test the method out on a couple of real-world datasets, where we may have some confidence in cell-type annotations so they could be used for benchmarking (e.g. ideally through FACS sorting but that could be hard to find, expert annotated data from a bio publication could be fine too) ? If possible, It would be additionally nice to see how large the discrepancy is between the quality of alignment with the self-tuned procedure vs a small grid of varying hyperparameters to see where in the range the self-tuned hyperparameters are performance-wise.
>
> **2. More of a concern than a question:** It appears that you don't consider $\rho_1$ and $\rho_2$ in the self-tuning procedure and fix them to 1.0. This works well in your practice because all the dataset you showcase your algorithm on are essentially balanced dataset; they have the same number of cells and same proportion of cell types because they came from experiments where measurements were jointly taken. This will not be the case for real-world applications and when datasets are unbalanced, $\rho$ values really matter based on my practical experience with unbalanced OT. I think testing your algorithm on some real-world separately sequenced datasets will likely show this.
>
> **3.** Shouldn't the threshold value eta be dependent on size of T matrix (number of samples)?
>
> **4.** Are the runtimes reported in the table on page 22 for a single combination of hyperparameter? I ask because $\epsilon$ updates in Algorithm 2 are going to be super small initially. For a starting value of $\epsilon=10^{-4}$, the next 10 $\epsilon$ values would be: 0.00010000005, 0.0001000001, 0.00010000015, 0.0001000002, 0.00010000025, 0.0001000003, 0.00010000035, 0.0001000004, 0.00010000045, 0.0001000005, 0.00010000055. These are tiny increments and for each update, the whole spectral wavelet GWOT is run again. This seems like quite a computationally heavy procedure and could prohibit the adoption of the method in real-world applications. If you run your algorithm on a couple of real-world datasets, could you also report how long the self-tuning procedure takes for these?

---

> > ### Comment · Reviewer_1Hb6 · 2024-11-21
> > **Minor comment on Question 5**
> >
> > In my opinion, a more informative experiment to demonstrate the value of wavelets would have been to compare this approach against GW with another spectral / graph-based distance that doesn't involve wavelets. Euclidean distance is pretty much guaranteed to not be great for high-dimensional applications like this, even if dimensionality reduction is used because in single-cell datasets, people typically use more than a couple PCs, for example, in order to capture majority of the variability. So in practice, I believe people use heat kernels or shortest path distances in nearest neighbor graphs in GW. Having said that, I have personally been more focused on the non-simulated data applications.

---

> > > ### Author Response · Authors · 2024-11-22
> > >
> > > Thank you for your continued engagement in our work. We answer your questions below:
> > > ***
> > > **Questions 1 & 2**
> > >
> > > Having real-world separately sequenced datasets is indeed an important consideration. While time constraints of the rebuttal period prevent us from conducting the full suggested experiment, we instead performed a focused analysis to address the core concern about performance in non-1-to-1 mapping scenarios.
> > >
> > > Specifically, we tested our method on simulated cell-type imbalance and missing cell types in the SNARE-seq and scGEM datasets. This simplified experiment allowed us to evaluate how our self-tuning procedure performs when the underlying correspondence is not strictly 1-to-1. We followed the experimental setup described in section 3.1.2 of Demetci et al. [1], where they simulated unbalanced single-cell datasets. We matched all hyperparameters (including $\rho$) used in [1], except for the epsilon regularization term, which is determined by Algorithm 2.
> > >
> > > The results of this new experiment are presented in the table below (note that all baseline results are taken directly from Table 1 in [1]):
> > >
> > > | Label Transfer Accuracy | SNARE (missing cell-type) | SNARE (subsam. cell-type) | scGEM (missing cell-type) | scGEM (subsam. cell-type) |
> > > |-------------------------|----------------------------|----------------------------|----------------------------|----------------------------|
> > > | E-WOT (heat kernel)     | 0.722                      | 0.806                      | **0.670**                      | 0.563                      |
> > > | E-WOT (simple tight)    | 0.684                      | 0.803                      | 0.661                      | 0.569                      |
> > > | L-WOT (heat kernel)     | **0.924**                      | 0.775                      | 0.596                      | 0.503                      |
> > > | L-WOT (simple tight)    | 0.691                      | **0.842**                      | 0.624                      | **0.642**                      |
> > > | SCOTy2                  | 0.653                      | 0.751                      | 0.521                      | 0.415                      |
> > > | SCOT                    | 0.572                      | 0.588                      | 0.323                      | 0.314                      |
> > > | Pamona                  | 0.423                      | 0.419                      | 0.414                      | 0.308                      |
> > > | MMD-MA                  | 0.407                      | 0.431                      | 0.296                      | 0.287                      |
> > > | UnionCom                | 0.406                      | 0.422                      | 0.315                      | 0.276                      |
> > > | bindSC                  | 0.584                      | 0.475                      | 0.254                      | 0.262                      |
> > > | Seurat                  | 0.477                      | 0.428                      | 0.377                      | 0.329                      |
> > >
> > > The results demonstrate that WOT is not only capable of handling unbalanced datasets but also surpasses the performance of current state-of-the-art methods for aligning unbalanced single-cell datasets.
> > >
> > > However, we want to clarify that the primary focus and novelty of this paper is in the underlying framework of WOT where the balanced formulation is most explored. While we have shown the capability of WOT to handle unbalanced datasets, we do not claim to have thoroughly explored all the edge cases of the unbalanced formulation in this work. We believe that a comprehensive analysis and evaluation of the unbalanced version of WOT on unbalanced datasets is beyond the technical scope of this paper. We acknowledge that further research is needed to fully investigate the performance and behavior of the unbalanced formulation.
> > >
> > > ***
> > > **Question 3**
> > >
> > > You are correct that the threshold η should ideally scale with the size of the T matrix to account for varying sample sizes. In our current experiments, we used a fixed threshold independent of matrix size, which is a limitation of our current implementation.
> > > ***

---

> > > > ### Author Response · Authors · 2024-11-22
> > > >
> > > > (continued)
> > > > ***
> > > > **Question 4**
> > > >
> > > > We should clarify two important points about our implementation:
> > > > 1. The ε updating scheme in our implementation uses a geometric progression (0.0001, 0.0005, 0.001, 0.005, 0.01, 0.05, 0.1, 0.5, 1.0) rather than the arithmetic progression described in the question. This significantly reduces the number of iterations needed to explore the meaningful range of ε values.
> > > > 2. The runtimes reported are indeed for a single hyperparameter combination, as our primary objective was to demonstrate the computational efficiency of the core WOT algorithm itself. We acknowledge that we should have been more explicit about this in our presentation.
> > > >
> > > > While hyperparameter selection is an important practical consideration, we focused our computational analysis on the WOT algorithm since it represents our main theoretical contribution. The current hyperparameter selection procedure is admittedly heuristic and was not optimized for computational efficiency, as our primary goal was to handle invalid transport plans. We recognize that for practical applications, more efficient hyperparameter selection strategies could be developed.
> > > >
> > > > We have added a note in the paper to clarify these points and to acknowledge that the total computational cost including hyperparameter selection would be higher than the reported single-run times.
> > > > ***
> > > > **Minor Comment on Question 5**
> > > >
> > > > We want to clarify our experimental design rationale: we specifically chose Euclidean distance as our baseline to isolate and demonstrate the noise-reduction capabilities of wavelets themselves. While we agree that alternatives like heat kernels or shortest path distances in nearest neighbor graphs often perform better in practice for single-cell applications, these methods already incorporate their own noise-reduction properties.
> > > >
> > > > Our bifurcation experiment was deliberately designed to show that wavelets provide meaningful improvements even when using simple Euclidean distances as the affinity matrix. This helps demonstrate the fundamental value of the wavelet approach independent of other noise-reduction techniques. Importantly, we view our wavelet-based method as complementary to, rather than competing with, distance metrics like shortest path distances. In fact, for all later experiments, we use the shortest paths to construct the affinity matrix for WOT (and for baselines).
> > > > ***
> > > > [1] Scotv2: Single-cell multiomic alignment with disproportionate cell-type representation. Journal of Computational Biology, 2022.

---

### Author Response · Authors · 2024-11-18
**Response to Common Questions & Concerns**

We thank the reviewers for their time, effort, and constructive feedback. We address some common questions and concerns brought up by multiple reviewers:
***
**Analyzing Filters and Wavelets Scales of E-WOT and L-WOT (Reviewer oyEN, Txuv)**

We have **added a new section (Appendix D.2.2)** that analyzes the wavelet scales of the single-cell datasets and filters of E-WOT and L-WOT. This section provides further insight into which wavelet scales are informative and why filters based on L-WOT or E-WOT may perform better in different single-cell datasets.
***
**Hyperparameter Selection and Many Tunable Hyperparameters (Reviewer 1Hb6, Txuv)**

While we acknowledge that the range of flexibility with our framework can be initially daunting, we want to make clear that most hyperparameters can be fixed to default values as specified in Table 3 of Appendix C. There are only three hyperparameters that are variable between each experiment: (1) the $\epsilon$ regularization, which is a hyperparameter common to any entropic OT method, (2) the aggregation operation, and (3) the weight normalization of the graph edges.

We **added Algorithm 2 to Appendix C**, which summarizes the unsupervised hyperparameter procedure originally described at the beginning of Appendix C in our initial submission.

It is worth noting that the challenge of understanding hyperparameter interactions is inherent to most modern machine learning methods. For instance, neural networks require tuning of learning rates, batch sizes, learning rate schedulers, weight initialization schemes, layer widths, and gradient clipping, among others. Similarly, different optimal transport variants (entropic, unbalanced, etc.) each introduce their own sets of hyperparameters. While we aim to provide clear guidance where possible, WOT's parameter space is comparable to many standard machine learning approaches.
***
**General Applicability of WOT (Reviewer tWjN, Txuv)**

Indeed, WOT can be applied to general unpaired dataset alignment problems, which we view as a strength rather than a limitation. Many successful methods in machine learning, such as Gromov-Wasserstein optimal transport, were initially designed for specific applications (e.g., object matching) but found wide applicability across various domains.

While WOT is a general solution, it was specifically inspired by and tailored to address challenges in single-cell data analysis:

* High dimensionality: Single-cell data often has thousands of features, requiring methods that can efficiently handle high-dimensional spaces. WOT's approach is well-suited for this task, unlike many shape-matching methods that can only operate on 3 dimensions [2, 3].
* Non-isometry: Unlike many natural language processing or computer vision [4, 5] tasks that often assume isometry, single-cell data from different modalities (e.g., RNA-seq vs. ATAC-seq) can have fundamentally different structures. WOT's multi-scale approach elucidates the similarities while filtering the dissimilarities between single-cell modalities.
* Noise and dropout: Single-cell data is notoriously noisy and sparse due to technical limitations in data acquisition. WOT's filtering mechanisms (both entropy-based and learned) are designed to mitigate these issues.

While these characteristics are not unique to single-cell data, their combination and severity in this domain motivated our approach. Our extensive experiments on single-cell datasets (SNARE-seq and scGEM) demonstrate WOT's effectiveness in this specific biological context.

That said, we agree that WOT's applicability extends beyond single-cell biology. This broader applicability is, in fact, a significant strength of our method.
***
[1] SCOT: single-cell multi-omics alignment with optimal transport. Journal of computational biology, 2022.

[2] NeuroMorph: Unsupervised Shape Interpolation and Correspondence in One Go. CVPR, 2021.

[3] NCP: Neural correspondence prior for effective unsupervised shape matching. NeurIPS, 2022.

[4] Gromov Wasserstein Alignment of Word Embedding Spaces. EMNLP, 2018.

[5] Cross-Lingual Alignment of Contextual Word Embeddings, with Applications to Zero-shot Dependency Parsing. NAACL, 2019.

---

### Meta-Review · Area_Chair_wRpQ · 2024-12-17

**Metareview:**

The paper proposes Wavelet Optimal Transport (WOT) for aligning unpaired single-cell datasets by leveraging spectral graph wavelets to address challenges like noise, dropout, and non-isometry. While the method offers a novel approach with multi-resolution alignment, the reviewers highlighted multiple weaknesses. These include unclear presentation, limited comparisons to key baselines like Harmony and scDML, and insufficient experimental validation on diverse real-world datasets. Reviewers emphasized that, while the work is non-trivial and technically valid, it lacks broader validation and deeper theoretical insights, reducing its appeal to the broader community. Furthermore, concerns were raised regarding the fixed hyperparameter settings, computational costs, and the lack of evaluation for multiple dataset alignments. Despite the authors’ efforts in addressing some of these points in the rebuttal, significant gaps remain in experimental breadth and theoretical depth, weakening the overall contribution. Given these limitations, the paper does not yet meet the standards for ICLR.

**Additional Comments On Reviewer Discussion:**

During the rebuttal period, the discussion focused on the lack of experimental diversity, hyperparameter sensitivity, and unclear practical applications. The authors responded with clarifications and incremental updates, such as analyzing filter scales and providing hyperparameter tuning details. However, these changes were not sufficient to address the broader concerns. The reviewers’ consensus weighed heavily on the need for broader applicability and deeper insights, ultimately leading to the decision to reject the paper.

---

### Decision · Program_Chairs · 2025-01-22

Reject